# ARTseq-FISH reveals position-dependent differences in gene expression of micropatterned mESCs

Xinyu Hu [1,2,3], Bob van Sluijs[1,3], Óscar García-Blay [1,2], Yury Stepanov[1], Koen Rietrae[1], Wilhelm T. S. Huck [1] ✉ & Maike M. K. Hansen [1,2] ✉

Differences in gene-expression profiles between individual cells can give rise to distinct cell fate decisions. Yet how localisation on a micropattern impacts initial changes in mRNA, protein, and phosphoprotein abundance remains unclear. To identify the effect of cellular position on gene expression, we developed a scalable antibody and mRNA targeting sequential fluorescence in situ hybridisation (ARTseq-FISH) method capable of simultaneously profiling mRNAs, proteins, and phosphoproteins in single cells. We studied 67 (phospho-)protein and mRNA targets in individual mouse embryonic stem cells (mESCs) cultured on circular micropatterns. ARTseq-FISH reveals relative changes in both abundance and localisation of mRNAs and (phospho-)proteins during the first 48 hours of exit from pluripotency. We confirm these changes by conventional immunofluorescence and time-lapse microscopy. Chemical labelling, immunofluorescence, and single-cell time-lapse microscopy further show that cells closer to the edge of the micropattern exhibit increased proliferation compared to cells at the centre. Together these data suggest that while gene expression is still highly heterogeneous position-dependent differences in mRNA and protein levels emerge as early as 12 hours after LIF withdrawal.

A complex interplay of signalling pathways and transcriptional networks guides morphological transformations during mammalian development[1–3]. During early development, the balance between key signalling pathways defines the cellular choice between differentiation and self-renewal[4–6]. Therefore, early changes in mRNA and protein expression levels can influence cell fate decisions[7–9]. Micropatterning has proven to be a powerful tool in unveiling position-dependent differentiation that recapitulates the spatial patterning of germ layers observed in early embryonic development[10–17]. Measuring mRNA and protein levels as mouse embryonic stem cells (mESCs) exit pluripotency will offer insights into how micropattern positioning affects early gene expression changes. This requires a scalable image-based approach capable of simultaneously detecting and quantifying mRNAs and proteins with a high spatial resolution across large areas.

Recently, next-generation sequencing (NGS)-based spatial-omics methods have greatly increased the multiplexing capacities of RNA or protein detection, albeit with low efficiency due to in situ reverse transcription and limited resolution of the beads used to capture targets[18–22]. Techniques that resolve the resolution issue, predominantly for RNA, are image-based spatial-omics methods. Image-based transcriptomic methods have greatly expanded the number of transcripts that can be resolved, using sequential fluorescence in situ hybridisation (FISH)[23–27] and barcoding approaches[24–27]. These methods have been combined with immunofluorescence (IF), but multiplexed protein detection remains a challenge[28]. For example, targeting and visualising RNAs and proteins often require separate sequential experimental and imaging steps, doubling the experimental time[26]. Thus, scaling of these techniques in terms of multiplexing capabilities

[1]Institute for Molecules and Materials, Radboud University, Heyendaalseweg 135, 6525 AJ Nijmegen, the Netherlands. [2]Oncode Institute, Nijmegen, The Netherlands. [3]These authors contributed equally: Xinyu Hu, Bob van Sluijs. ✉e-mail: w.huck@science.ru.nl; maike.hansen@ru.nl

are bottlenecks, limiting the ability to simultaneously visualise mRNAs and proteins in high throughput[29–32]. Furthermore, the resolution of protein detection is often below the level of RNA detection, proving a barrier to quantitative insights into protein location or spatial correlation of RNA and protein abundance[24,26,28–32]. Conversely, DNA-barcoding dependent methods (for example, CODEX[33], ImmunoSABER[34] and InSituPlex®[35]) have shown their multiplexing capacity in protein detection, but have thus far not been integrated with RNA detection[36,37].

Here we develop and validate a multiplexed spatial profiling method: antibody-mRNA targeted sequential FISH (ARTseq-FISH). This method enables simultaneous quantification of mRNA, protein, and phosphoprotein levels in individual mESCs with high spatial resolution. The gene expression profiles of 67 targets were analysed during the initial 48 hours of differentiation, allowing us to capture relative changes in mRNA and protein abundance as mESCs exit pluripotency. Similar to previous evidence[15,16], ARTseq-FISH reveals distinct gene-expression profiles in cells located at different positions on the micropattern. Specifically, position-dependent differences in gene expression profiles emerge already during the first 12 h of Leukaemia inhibitory factor (LIF) withdrawal. Finally, we confirm that cells closer to the edge of the micropattern display increased proliferation compared to cells at the centre of the micropattern already during the first 12 h of LIF withdrawal. In summary, we describe and validate a method that quantifies relative differences in mRNA and (phospho)protein abundance.

## Results

### Simultaneous visualisation of mRNAs and (phosphor)proteins

The ARTseq-FISH workflow comprises a series of consecutive steps (Fig. 1a, d, Supplementary Note 1 and Supplementary Figs. 1–16). In short, fixed cells are stained with DNA-barcoded antibodies against (phospho-)protein targets of choice (Supplementary Data 1, 2). The DNA barcode of each antibody, known not to influence antibody binding efficiency[38,39], contains a 10 nt polyadenylation (poly-A) linker and a specific 36 nt sequence that is complementary to a corresponding target-specific single-stranded DNA (ssDNA) padlock probes (PLPs). On the other hand, cellular mRNAs are directly targeted by one

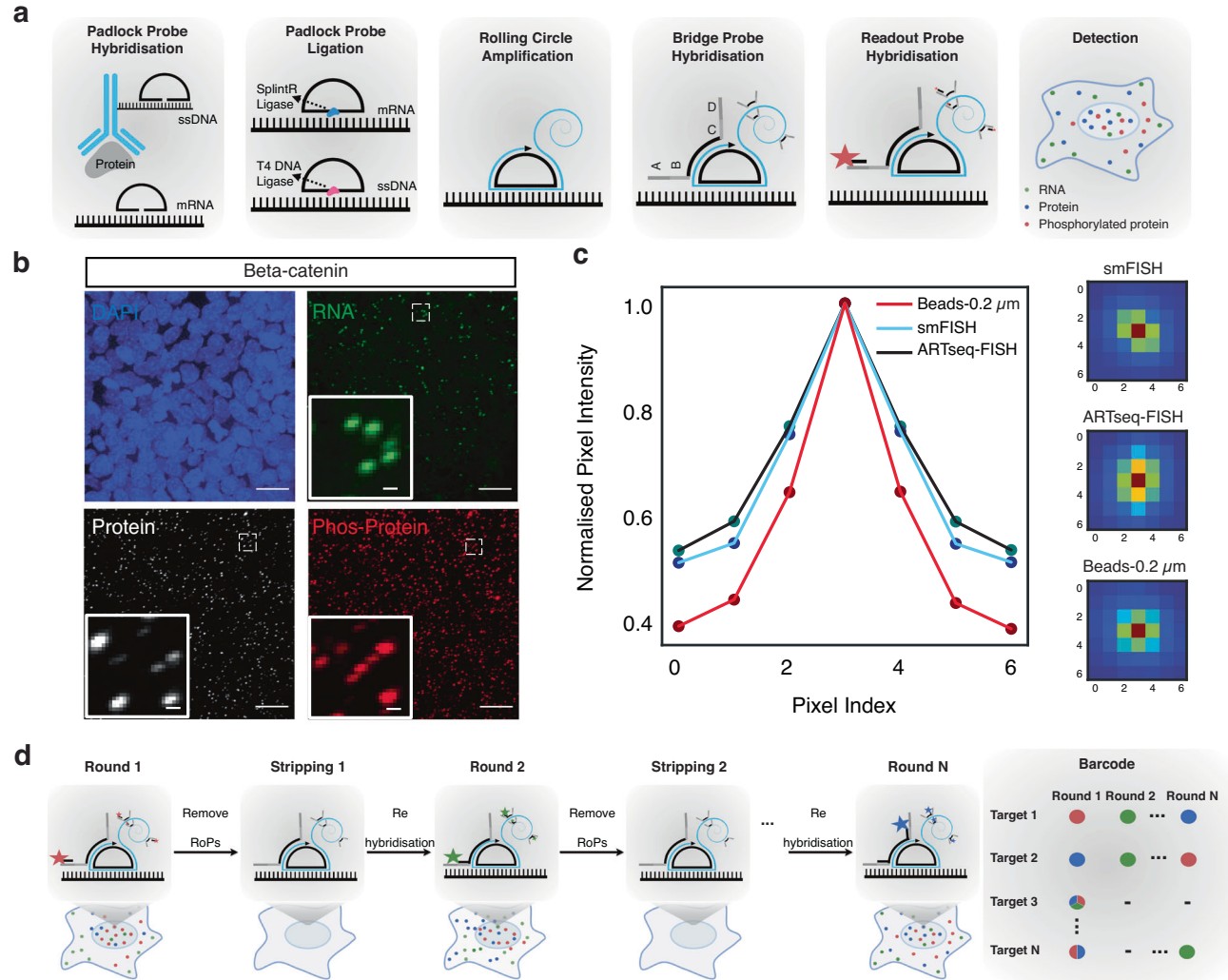

**Fig. 1 | Experimental outline of ARTseq-FISH. a** The experimental workflow of ARTseq-FISH enables the detection of different molecular species (mRNA, proteins and phosphoproteins) simultaneously. Each antibody conjugated to oligonucleotides, targets its associated (phospho-)protein in situ. Target-specific PLPs bind to these oligonucleotides as well as cellular mRNA, respectively. Then the PLPs are circularised and amplified into RCPs. A set of target-specific BrPs hybridise to RCPs, and then RoPs labelled with fluorophores bind to one of the four flanking sequences on the BrP. **b** Representative raw data of simultaneously detecting *Beta-catenin* mRNA, protein, and phosphoprotein with the same resolution. Scale bar, 20 μm, 1 μm. **c** The relative mean pixel intensity of the hybridisation signal's point spread functions (*n* = 2000) around the local maxima. Comparison of the average observed signal of ARTseq-FISH to smFISH and a 200 nm fluorescent bead. **d** The sequential stripping and rehybridisation of new known RoPs allows subsequent decoding of stacked signals. Source data are provided as Source Data sheet tab Fig. 1c.

specific ssDNA PLP with a target sequence of 36 nt (Supplementary Fig. 1a, b), resulting in DNA/RNA hybridisation. Therefore, the circularising of the PLPs that hybridise to either proteins or RNAs require T4 DNA ligase and SplintR ligase, respectively[40–42] (Fig. 1a and Supplementary Figs. 4–6). Next, rolling circle amplification (RCA) generates rolling circle products (RCPs) that amplify the original (phospho-) protein or mRNA target (Fig. 1a and Supplementary Fig. 8). To decode the original target amplified by the RCPs, we first hybridise target-specific bridge probes (BrPs), which are each recognised by unique combinations of fluorescently labelled readout probes (RoPs) (Fig. 1a, Supplementary Figs. 1c, d, 9 and Supplementary Data 1), allowing the detection of individual spots for each mRNA and protein (Fig. 1b, c and Supplementary Figs. 17, 18). ARTseq-FISH provides similar relative protein abundance compared to classic immunofluorescence (Supplementary Fig. 19). We determined a detection efficiency of ~47% (for *Nanog* mRNA) by comparing mRNA quantification using ARTseq-FISH with conventional single molecule RNA FISH (smFISH)[27] (Supplementary Fig. 20 and Supplementary Data 3). Furthermore, ARTseq-FISH enhances the signal ~4-fold with respect to conventional smFISH (Supplementary Fig. 20c, d), resulting in a higher signal-to-noise ratio. On average, ~0.03–0.5 spots per cell were detected as false positives from unspecific PLP (protein detection only), BrP, and RoP hybridisation (Supplementary Fig. 3), in particular in the presence of true hybridisation signal (Supplementary Fig. 18b). Similar to existing methods, we are unable to quantify the degree of false positives coming from unspecific binding of the antibody or PLP, and the target detection efficiency is highly antibody[33,34] and PLP-dependent[40–43] (Supplementary Fig. 10). Lastly, we specifically employ one PLP per mRNA target, since we are interested in relative mRNA differences. However, should more rigorous quantification of mRNA be necessary, at least three PLPs per target are likely required[40–43].

## Automated quantification of mRNAs and proteins in cells

Multiplexed, simultaneous detection and quantification of mRNAs and (phospho-)proteins over a large area is achieved through serial stripping and rehybridisation of RoPs following a colour barcoding scheme[27] (Fig. 1d, Methods, Supplementary Notes 1, 2, Supplementary Figs. 14–16 and Supplementary Data 4). The barcode of each target is determined by the presence of a given colour (green, red, or far red) in a sequence of specific hybridisation rounds. To correctly decode the detected signal and translate this into mRNA or protein counts for individual cells, we developed an automated image analysis pipeline that returns the abundance of each target per cell.

The pipeline (Fig. 2a, more detailed description in Supplementary Notes 2, SN2 Figs. 1–3, software packages used[44–49]) initiates the analysis by identifying the local maxima per colour channel[50] by denoising and sharpening the image in six steps (Supplementary Notes 2, SN2 Figs. 3–4) and classifying the local maxima as a true hybridisation signal. This is done by using a support vector machine (SVM) model trained on data of true point spread functions, which allows for signal deconvolution (Supplementary Notes 2, SN2 Figs. 5–6). Single-cell analysis requires the reconstruction of single cells in 3D from a collection of images (Fig. 2a and Supplementary Notes 2, SN2 Figs. 7–15). However, consistent segmentation is challenging because a segmentation algorithm that maps onto one nucleus might not map onto others (Supplementary Notes 2, SN2 Fig. 8), or a global image filter that cleanly separates the nuclei in one image might not be optimal for another (Supplementary Notes 2, SN2 Fig. 10). Therefore, an iterative approach was implemented, repeatedly segmenting nuclei using different global image filters and segmentation parameters[51] (Supplementary Notes 2, SN2 Figs. 9 and 13). We collected the segmentation RoP labels and used an SVM model to filter out any failed segmentations. The remaining labels were subsequently used to reconstruct the nuclei in 3D by clustering their centroids with a node-based graph (Supplementary Notes 2, SN2 Figs. 14, 15). After interpolating any missing nucleus segmentations across the z-axis, we obtained a 3D bounding box. Another SVM model subsequently assessed if this bounding box contained a single nucleus. Before assigning mRNA and protein counts to a cell, we adjusted for any drift, by using the segmented nuclei as unique markers to align the images. Once a global shift was found, we refined the alignment by moving all hybridisation signals across a narrow range of shifted pixels until the signals overlapped (Supplementary Notes 2, SN2 Figs. 16, 17). Ultimately, we decoded the overlapping signals and assigned each hybridisation signal to a particular protein or mRNA in a specific cell (Fig. 2b and Supplementary Fig. 2, Supplementary Notes 2, SN2 Figs. 18, 19). Cytoplasmic targets were assigned to a specific cell based on the shortest distance to the nucleus (Supplementary Notes 2, SN2 Fig. 23).

## Expression profiles of mRNAs and (phospho)proteins in mESCs

We utilised ARTseq-FISH to explore how the expression of mRNAs and (phospho-)proteins is affected during the early stages of differentiation on micropatterns. LIF/serum cultured mESCs are heterogenous (Supplementary Fig. 22), due to their pluripotent nature, enabling them to make different cell fate decisions upon LIF withdrawal[52–54]. Therefore, we plated mESCs directly on a 750-μm-diameter micropattern and withdrew LIF from the serum medium for up to 48 h to allow for cells to exit pluripotency. Notably, cells cultured under this condition are not the same as the cells primed to epiblast-like cells (EpiLCs)[12]. After fixation, we performed ARTseq-FISH on the micropatterned mESCs. Using our analysis pipeline, we visualised the locations of unique mRNA and (phospho-)protein targets (Fig. 3a) and quantified the abundance of 67 targets in individual cells (Fig. 3a–d). When discussing the protein targets detected using the respective antibodies (Supplementary Data 2), it is important to note that these measurements represent the total amount of protein detected, which includes both the unmodified and modified forms. Conversely, phosphorylated proteins were detected with specific antibodies (*protein or phos-protein in figures).

Among protein targets, lineage-associated markers i.e. N-Cadherin, FOXA2, GATA6, showed higher abundance than pluripotency-related markers, i.e. NANOG, OCT4 and SOX2. Although we cannot rule out that these differences are due to dissimilarities in antibody efficiency, this is expected in mESCs that are exiting pluripotency (Fig. 3b)[10,12,15]. Notably, while RB shows low spot counts at a protein level, its mRNA and Phos-RB display higher spot counts (Fig. 3b–d), indicating that the difference in spot count between total RB and Phos-RB is caused by differences in antibody specificity. As mentioned previously, similar to other omics methods that rely on antibodies[33,34], ARTseq-FISH is dependent on the binding efficiency of primary antibodies (for proteins) and PLPs (for mRNAs). Therefore, we cannot compare absolute counts between species. However, strong correlation of target counts between four different biological replicates (Fig. 3f) demonstrates the high reproducibility of ARTseq-FISH (Supplementary Note 2 and Supplementary Fig. 21).

In addition, after filtering out the low abundant RNA targets in single-cell RNA sequencing (scRNAseq) data, ARTseq-FISH and scRNAseq mRNA quantification showed a high correlation ($r^2 = 0.73$) (Fig. 3g). Finally, to ensure true signals were detected with the image analysis software since no fixed intensity threshold is implemented, we tested the software on three samples: (i) samples that did not include any readout probes, i.e. negative control (Fig. 3e, top panel); (ii) samples that included readout probes, i.e. true signal; and (iii) an in silico merged image which combined samples i and ii. The ATTO 488 channel was the noisiest channel. For the merged images with ATTO 488 the number of false positives decreased at minimum sixfold (Fig. 3e, bottom panel). This demonstrates that in the presence of true signal, the false positive rate for the noisiest channel is <1%. Together

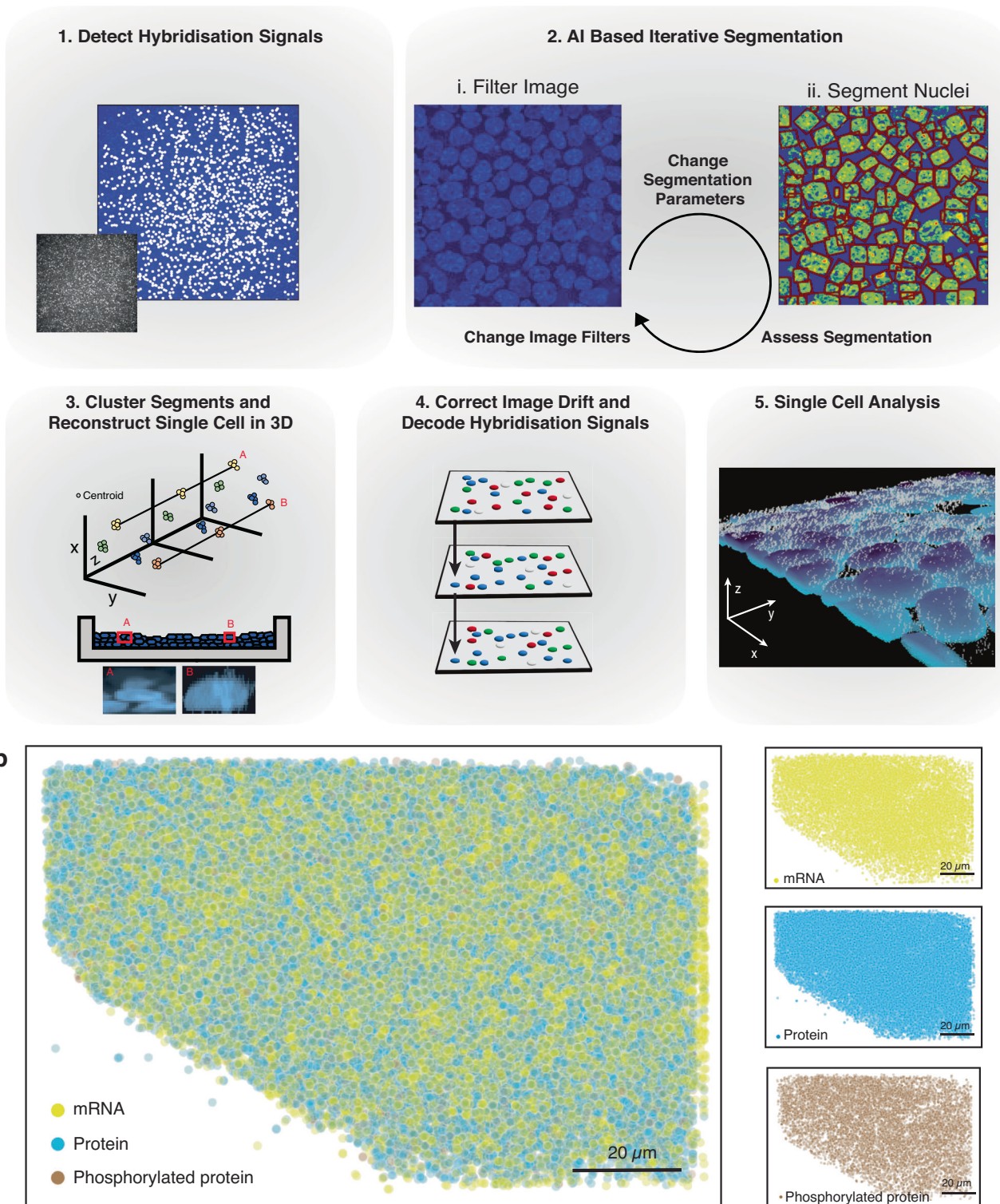

**Fig. 2 | Computational outline of ARTseq-FISH. a** A condensed summary of the automated computational pipeline that converts the raw microscope images into a single-cell dataset, providing the abundance of each target per cell. First, the hybridisation signal is detected, followed by an iterative AI-based segmentation approach, the segments are clustered, and the cells are reconstructed in 3D. Next, an image drift correction is applied to align the hybridisation between rounds. In the final step, these hybridisation signals are decoded and assigned to the targets as well as individual cells. **b** Reconstructed images show all the detected molecules in the cells cultured on micropattern 48 h after leukaemia inhibitory factor (LIF) withdrawal. The spots are subdivided into three classes: mRNA (yellow), protein (blue) and phosphorylated protein (brown). Scale bar, 20 μm.

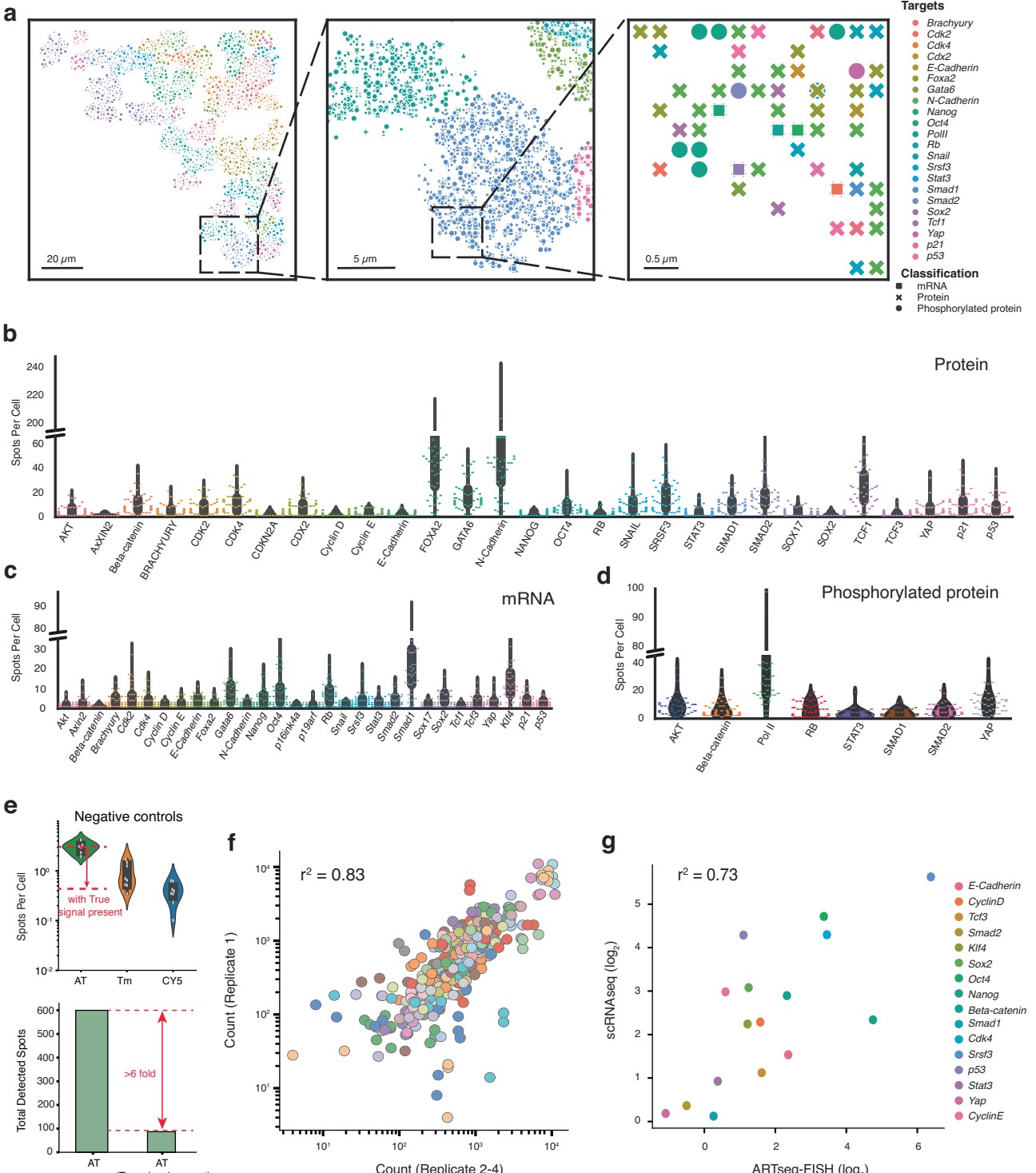

**Fig. 3 | ARTseq-FISH profiles the distribution of mRNAs and proteins in individual mESCs on micropatterns. a** Reconstructed images show the detected spots in Fig. 2b assigned to individual cells and the identity of these targets at a specific location. Scale bar, 20 μm, 5 μm and 0.5 μm. **b–d** Quantification of proteins (**b**), mRNAs (**c**) and phosphorylated proteins in single cells (**d**). Forty-eight cells were analysed per target. **e** Quantification of spots per cell for negative controls in three different channels (the three fluorophores are AT: ATTO 488, Tm: TAMRA, and CY5). Negative controls were performed without PLPs and 579 cells were analysed. When adding true signal to negative control images in silico for the ATTO 488 channel, the number of detected spots decreased >6-fold (shown; merged image

with the smallest fold change reduction in false positives) for the green channel (bottom). **f** The scatter plot shows the average correlation between the total count of individual targets detected within the image for four biological replicates at 48 h differentiation of mESCs. **g** The correlation of RNA targets per cell between ARTseq-FISH and scRNAseq performed on serum/LIF cultured mESCs. scRNAseq data were analysed by Seurat[88]. 455, 503, 470, 482, 454, 466, 488, 408, 408, 408, 180, 241, 200, 200 and 233 non-micropatterned cells were analysed for *Smad2, Tcf3, E-cadherin, Cdk4, CyclinD, Smad1, Klf4, Oct4, Nanog, Sox2, p53, Beta-catenin, Stat3, CyclinE, Yap* respectively. Source data are provided as Source Data sheet tabs Fig. 3a–g.

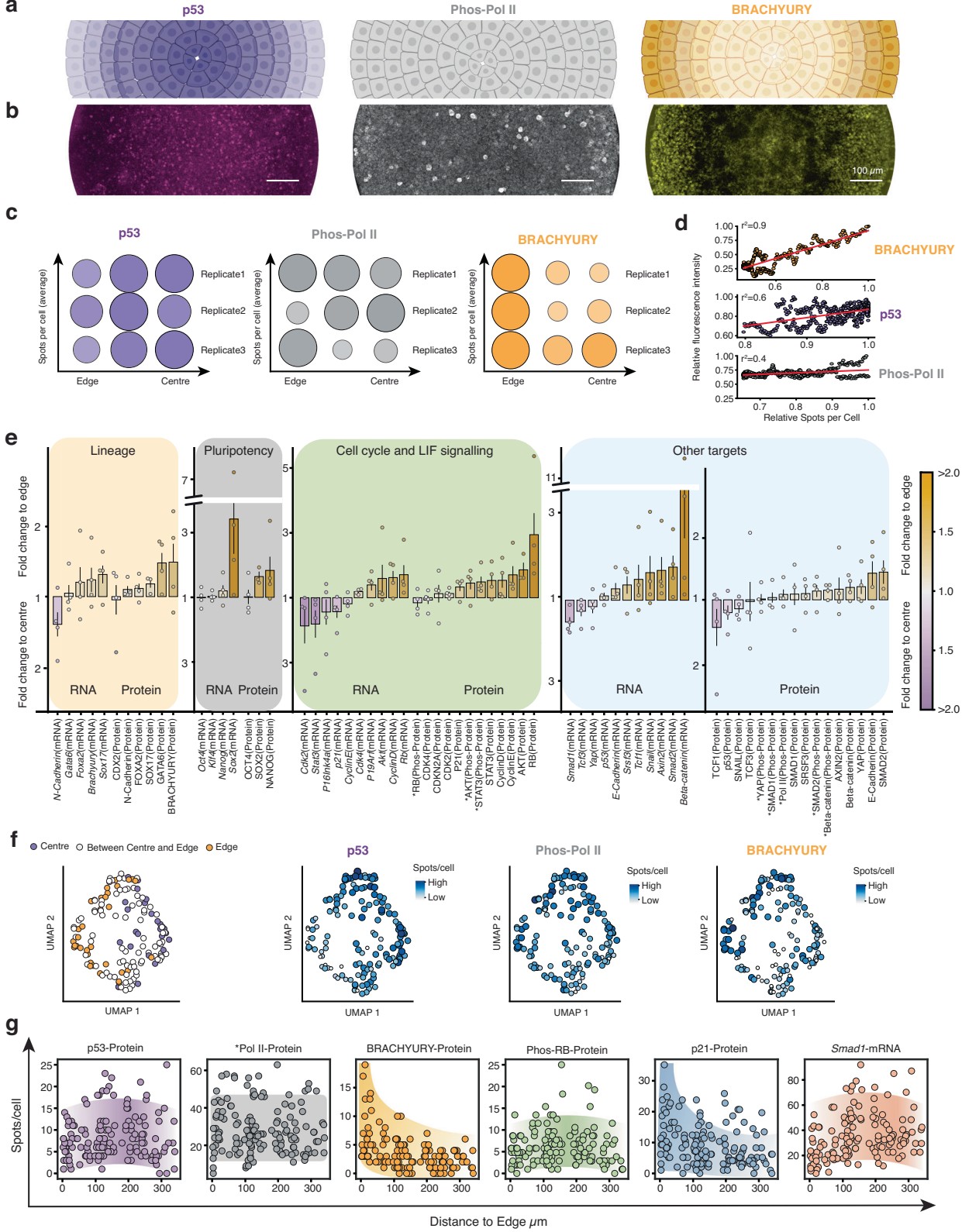

these data indicate that ARTseq-FISH is a reproducible technique that should be implemented when comparing relative spatial or temporal changes in expression levels of mRNA and proteins.

## Expression gradients across micropatterned mESCs

ARTseq-FISH provides single-cell quantification (Figs. 1–3) of each molecular species while retaining spatial information. As confirmed by immunofluorescence, ARTseq-FISH data show three patterns of molecular distribution across mESCs on micropatterns: (i) increased expression in cells located at the centre (i.e. p53); (ii) evenly distributed throughout (i.e., phosphorylated RNA Pol II); and (iii) increased expressed in cells at the periphery (i.e. BRACHYURY) (Fig. 4a–f and Supplementary Fig. 23). Consistently, targets are classified into these three groups (Fig. 4e and Supplementary Fig. 24, purple, white, and

**Fig. 4 | ARTseq-FISH reveals gradients of mRNA and protein expression levels within micropattern-differentiated mESCs. a** Schematic illustration of three representative spatial distributions of molecules on micropatterns. Created with BioRender.com. **b** Immunofluorescence results of p53, Phos-Pol II (Phosphorylated RNA Pol II), and BRACHYURY within a micropattern-differentiated mESCs. Scale bar, 100 μm. **c** Average number of spots per cell at different positions on the micropattern (edge to centre) for p53, Phos-Pol II and BRACHYURY proteins from three biological replicates (150, 125 and 158 cells and 1 micropattern were per replicate). **d** Pearson correlation between relative mean fluorescence in (**b**) and relative spots per cell with respect to their position on the micropattern (**c**). **e** Mean fold change of the expression level of individual targets between the edge and centre (purple and orange respectively) of the micropattern across four biological

replicates (125–158 cells and 1 micropattern analysed per replicate). *: phosphorylated protein. Error bars represent the SEM of four biological replicates. **f** UMAP analysis of expression levels of 57 markers of individual cells after 48 h of LIF withdrawal, with the colour representing if cells are localised towards the edge (orange) or centre (purple) of the micropattern (left). The same UMAP with single cell abundance of p53, Phos-Pol II and BRACHYURY proteins indicated in blue. 158 cells and 1 micropattern were analysed. **g** The spots per cell (y-axis) with respect to the position of the cell on the micropattern (x-axis) of selected targets. The edge of the micropattern serves as the reference value for the relative position of the cells. Cells are found at the edge at 0 μm and near the centre at 375 μm. 158 cells were analysed per target. Source data are provided as Source Data sheet tabs Fig. 4c–g.

orange respectively). Overall, lineage, pluripotency as well as cell cycle associated proteins are more highly expressed in cells localised towards the edge of the micropattern. Specifically, BRACHYURY and GATA6 are mostly enriched in cells at the edge of the micropattern, as are the NANOG and SOX2 (Fig. 4e and Supplementary Fig. 25a). Yet, BRACHYURY and GATA6 are not necessarily highly expressed in the same cells as NANOG and SOX2 (Supplementary Fig. 25e). Surprisingly, some mRNAs (i.e. *Stat3* and *Cdk2*) appear localised to the centre although their (phospho-)protein counterparts are localised to the edge. This could either be because cells at the edge are translationally more active and thereby require less mRNA to produce a higher amount of protein. Alternatively, cells could be in the process of up- or downregulating their mRNA, and differences in mRNA and protein half-life could lead to a lack of correlation between the two species[55]. Lastly, when calculating the mean fold change in expression level in cells located in the centre versus edge, cells >125 μm from the edge were defined as being in the centre of the micropattern (Supplementary Fig. 24), which is possibly an oversimplification. For example, *Smad2* and *Rb* mRNA and protein are enriched in cells at the edge, while our analysis suggests that the phosphorylated proteins are more homogeneously expressed across the micropattern (Fig. 4e). However, when quantifying the average number of spots per cell at seven positions across the micropattern, a reduction in Phos-RB and Phos-SMAD2 protein at the centre (-375 μm from the edge) of the micropattern becomes evident (Supplementary Fig. 25b–d). Consequently, the localisation analysis likely masks more subtle changes in gene expression between cells at the edge and centre of the micropattern.

To more closely examine the relationship between target expression levels and cellular location, we quantified the levels of some targets in individual cells and plotted these against cellular position on the micropattern. This analysis revealed that targets that are more highly expressed in cells at the edge (i.e. BRACHYURY), display more of an exponential trend (Fig. 4g), rather than linear. Conversely, some targets that are classified as localised towards the centre (i.e. Phos-RB) or the edge (i.e. Smad1 mRNA) appear to be most abundant in cells located -150 μm from the edge, rather than 375 μm from the edge, the latter being the true centre. Interestingly, the single-cell abundance of individual targets is highly heterogeneous (Fig. 4g). While this could be a technical consequence of ARTseq-FISH, the heterogeneity of p53, Phos-PolII, and BRACHYURY is also visible with regular immunofluorescence (Fig. 4b), and mRNA levels have previously been reported to become more heterogeneous as cells exit pluripotency[56]. Furthermore, we have shown that the mean gene expression values are reproducible across replicates when quantified by ARTseq-FISH (Fig. 3f). Therefore, this heterogeneity is likely biological and indicates that although cellular location on the micropattern does generate trends in gene expression profiles, both mRNA and protein synthesis is still heterogenous after 48 h of LIF withdrawal.

**Position-dependent expression profiles emerge over time**

To investigate how mRNA and protein abundance changes over time, we examined the spatial distribution of molecules in micropatterned

cells at different time points (0, 12, 24 and 48 h) following LIF withdrawal (Fig. 5a). To better understand the changes, we grouped targets into different categories (Supplementary Data 5) and quantified the relative changes in target abundance across the micropattern in 2D (Fig. 5b) over the 48-h LIF withdrawal period. At the initial time point (0 h), both RNA and protein levels across the different categories showed a relatively homogeneous distribution. Interestingly, we noticed a specific region, -150 μm from the edge of the micropattern, where mRNAs related to cell cycle class 2 (genes that promote cell cycle progression) were elevated (Fig. 5c, cell cycle class 2). Moreover, this region exhibited slightly increased levels of mRNAs and proteins related to (LIF) signalling at 0 h (Fig. 5c, signalling). While we considered that this might be an artefact of initial uneven cell seeding, cells were seeded on the micropattern 24 h before LIF withdrawal (i.e. at T = −24 h). Therefore, these subtle initial expression differences could be indicative of the first changes in position-based expression profiles. This is reinforced by two distinct areas on the micropattern emerging where both mRNA and protein levels tended to increase over time: these were found at positions 2 and 6 of the micropattern, located -150 μm from either side of the edge (Fig. 5b).

To further explore whether proteins are expressed differently based on cellular location on the micropattern, we quantified gene expression profiles for individual cells situated either at the true edge or at the true centre of the micropattern (Fig. 5b). Once again, we categorised targets based on their functions and plotted the relative single-cell changes over the 0, 12, 24 and 48 h of LIF withdrawal. At 0 h, cells at the micropattern's edge contain higher levels of proteins associated with cell cycle 1 (lengthening G1) than at the centre (Fig. 5d, 0 h cell cycle 1 top compared to bottom). Subsequently, at the edge after 12 h, we observed a decrease in proteins associated with cell cycle 1 (lengthening G1) and an increase in proteins related to cell cycle 2 (cell cycle progression) (Fig. 5d). These findings suggested that within the first -12 h of LIF withdrawal, cells at the edge alter the expression of genes associated with cell cycle progression, potentially leading to higher proliferation rates during this time. In contrast, at the micropattern's centre, the levels of cell cycle 1 (lengthening G1) increase already from 0–12 h, while cell cycle 2 genes (cell cycle progression) initially remain constant (Fig. 5d, bottom). These results indicated that the G1 phase of cells located at the centre might start increasing 0–12 h after LIF withdrawal.

When analysing genes associated with pluripotency and particular lineages, we only observed slightly different gene expression profiles at the centre compared to the edge of the micropattern. Proteins that exhibit the most significant differences in expression profiles are SOX17, BRACHYURY, (Fig. 5d, right) and NANOG (Fig. 5e, left). In particular, SOX17 levels increase mostly between 0–12 h at the edge of the micropattern, while at the centre, there is a more gradual increase over the course of 48 h of LIF withdrawal. BRACHYURY levels increase gradually at the edge of the micropattern from 12 to 48 h of LIF withdrawal and remain relatively constant at the centre. Lastly, NANOG levels increase from 0 to 12 h and decrease again from 12 to 24 h of LIF withdrawal at the edge of the micropattern. In contrast, at the centre of

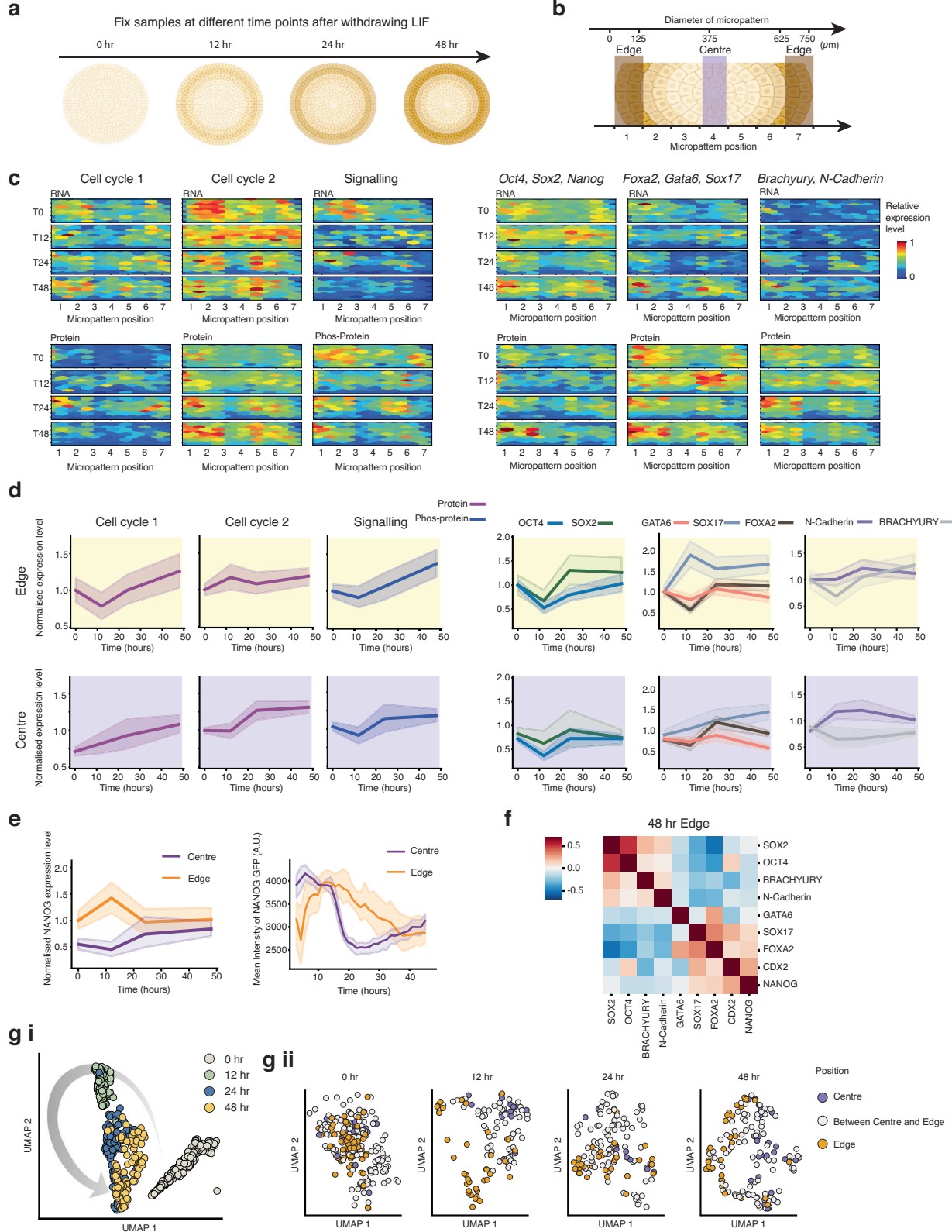

the micropattern, NANOG levels increase from 12 to 24 h of LIF withdrawal.

To validate these findings, we performed time-lapse microscopy on micropatterned NANOG-GFP mESCs[57,58] for 46.5 h after LIF withdrawal (Fig. 5e, right). When quantifying the mean NANOG-GFP intensity for different positions at the edge and the centre of the micropattern, we observed distinct NANOG-GFP expression changes.

Specifically, cells at the edge of the micropattern show an initial increase in NANOG-GFP expression (lasting ~12–15 h) followed by a gradual decrease in NANOG-GFP expression. Conversely, the centre of the micropattern displays a decrease in NANOG-GFP expression from 15 to 20 h post-LIF withdrawal, followed by a continuous increase in NANOG-GFP expression (Fig. 5e, right). While the overall trend at the edge is consistent with ARTseq-FISH NANOG data (Fig. 5e, left,

**Fig. 5 | Spatial-temporal quantification of mRNAs and proteins after LIF withdrawal from mESCs on micropatterns. a** Schematic of the experiment where ARTseq-FISH was performed at 0, 12, 24 and 48 h after LIF withdrawal. Created with BioRender.com. **b** Schematic of the definition of centre and edge on the micropattern. Created with BioRender.com. **c** Heatmap showing the abundance of different classes of targets (see Supplementary Data 5 for full list of targets) across 1 micropattern at 0, 12, 24 and 48 h after LIF withdrawal. The expression level of targets is normalised to the maximum abundance across all four time points. **d** Relative single-cell expression levels of the targets shown in (**c**), at the edge (top) and the centre (bottom) of the micropattern at 0, 12, 24 and 48 h after LIF withdrawal. 0–24-h data includes two biological replicates; 48 h includes three biological replicates (for the edge 393 cells and the centre 238 cells were analysed in total). For each replicate and time point, 1 micropattern was analysed. The line represents the mean. Shaded regions represent 95% confidence intervals. **e** Single cell NANOG protein expression at the edge and centre of micropattern at 0, 12, 24 and 48 h after LIF withdrawal detected by ARTseq-FISH (left). 0–24-h data includes

two biological replicates; 48 h includes three biological replicates (edge and centre include 393 and 238 cells in total, respectively). For each replicate and time point, 1 micropattern was analysed. Quantification of live cell time-lapse of NANOG-GFP positive mESCs for five different positions at the edge and the centre of 1 micropattern over 46.5 h of LIF withdrawal (right). The line represents the mean. Shaded regions represent 95% confidence intervals. **f** Pearson correlation and hierarchical clustering of pluripotency and lineage-related markers at the edge of the micropattern (3 biological replicates, 1 micropattern each, 111 cells total) at 48 h after LIF withdrawal. **g** (i) UMAP clustering of expression levels of 57 markers of individual cells within a single micropattern experiment at different time points (0, 12, 24 and 48 h) after LIF withdrawal. (ii) UMAP clustering of expression levels of 57 markers of single cells at different positions (centre, edge or between centre and edge) within the micropattern (purple, orange and white, respectively). One biological replicate and 1 micropattern (0, 12, 24 and 48 h include 269, 139, 141 and 158 cells, respectively). Source data are provided as Source Data sheet tabs Fig. 5d–g.

orange), the initial decrease observed in the time-lapse experiment for cells at the centre of the micropattern is not visible by ARTseq-FISH (Fig. 5e, left, purple). This could be because ARTseq-FISH reveals expression levels of cells that are fixed after 0, 12, 24 and 48 h post-LIF withdrawal and do not represent a true time course, which might conceal rapid changes in expression levels. Furthermore, the time-lapse experiment was not analysed at a single-cell level. Alternatively, it is possible that the differences between the time-lapse experiment and ARTseq-FISH are cell-type specific. Therefore, we sought to confirm the expression of another pluripotency marker (OCT4) by immuno-fluorescence at 0 and 48 h of LIF withdrawal, which shows similar behaviour as ARTseq-FISH (Supplementary Fig. 26).

Interestingly, at the single-cell level, we found a negative correlation between pluripotency-associated proteins (i.e. OCT4 and SOX2) and lineage-associated markers (i.e. FOXA2, GATA6 and SOX17) (Fig. 5f) in cells at the edge of the micropattern. This indicates that cells at the edge are not uniform in their protein expression levels. When we project the data on a 2D UMAP, we observe the most prominent separation of cellular expression profiles occurring between 0 and 12 h post-LIF withdrawal (Fig. 5g). Furthermore, while there is some separation in cells located at the edge and centre, this separation is subtle, and predominantly visible at 48 h post-LIF withdrawal (Fig. 5g).

## Cellular location on the micropattern impacts proliferation
In light of the gene expression profiles (Fig. 5c) indicating differences in protein signals between the micropattern's edge and centre, we proceeded to investigate the translational activity of cells across the micropattern. To do this, we exposed micropatterned mESCs to the puromycin analogue O-propargyl-puromycin (OPP), which is incorporated into newly synthesised proteins[59]. Through click chemistry, we fluorescently labelled the nascently translated polypeptides (Fig. 6a). Our data revealed a gradual decrease in protein synthesis across the entire micropattern over time (Fig. 6b). Notably, the signal in the micropattern's centre decreased more readily, particularly after 24–48 h (Fig. 6b, c, light and dark green). These findings are in line with ARTseq-FISH data, showing more prominent protein detection at the micropattern's edge after 48 h of LIF withdrawal (Fig. 6d). These data suggest that spatial organisation plays a role in the dynamics of translational activity in micropatterned mESCs.

To gain insights into why translation might be reduced in the centre of the micropattern, we next tried to determine if cells proliferate differently depending on their position on the micropattern. To this end, we first quantified cell density across the micropattern as a function of DAPI intensity. Interestingly, on our 750 μm diameter micropatterns, cell density shows two radially symmetric peaks over time, ~100 μm from the edge of the micropattern (Fig. 6e, f), indicating that the colonies are not flat but somewhat 'donut' shaped (Fig. 6g). To determine if these differences in cell density were caused by increased

proliferation, we performed EdU staining after 0, 12, 24 and 48 h of LIF withdrawal and quantified cell proliferation from the incorporation of EdU into newly synthesised DNA[60] (Fig. 6h). The results indicate that there are fewer cells in the S phase at the centre of the micropattern than towards the edge and therefore likely display reduced proliferation already after 12 h of LIF withdrawal (Fig. 6i). Interestingly, at the very edge of the micropattern there still appear to be cells incorporating EdU after 48 h of LIF withdrawal. To confirm that actively cycling cells are more abundant at the edge of the micropattern after LIF withdrawal, we performed Ki-67 staining[61] (Fig. 6j and Supplementary Fig. 27), an indirect marker for proliferating cells. The data show that at 0 h after LIF withdrawal, Ki-67 staining is mostly homogenous across the micropattern, indicating comparable cell cycling at the centre and edge of the micropattern (Fig. 6i). Notably, at 0 h there is reduced Ki-67 signal at the very centre of the micropattern (micropattern position 1 and 8), consistent with quantified EdU signal (Fig. 6i, light blue). From 0 to 12 h, there appears a slight increase in Ki-67 intensity (i.e. cell cycling) at the centre of the micropattern, which subsequently drops from 24 to 48 h post-LIF withdrawal, demonstrating a decrease in cellular proliferation or a state of cell cycle arrest (Fig. 6j and Supplementary Fig. 27). Caspase-3 and 7-AAD staining, as well as nuclear morphology analysis, indicate that this lack of proliferation could be connected to increased apoptosis at the centre of the micropattern[62] (Supplementary Fig. 28). Conversely, at the edge of the micropattern cells show more consistent Ki-67 intensities over time (Fig. 6j), although at 48 h the cells at the very edge (micropattern position 1 and 8), do not show increased cycling as would be expected from the EdU staining (Fig. 6j compared to Fig. 6i), this could be because Ki-67 is less sensitive, in particular for cells with a short G1 phase[63]. Nevertheless, these combined assays indicate that cells positioned more towards the edge of the micropattern proliferate more readily than cells at the centre.

## Cellular location on the micropattern impacts the cell cycle
Given the distinct expression profiles of cell cycle-related genes observed between cells at the micropattern's edge and centre using ARTseq-FISH (Fig. 5d), we aimed to quantitatively analyse cell cycle progression on a single-cell basis. To this end, we conducted time-lapse microscopy on micropatterned *Fucci* mESCs[64,65], which enabled the visualisation of cell cycle dynamics in live cells (Fig. 7a). Single-cell tracking and measurement of Azami Green (Az1) and Kusabira Orange-2 (KO-2) fluorescence over time delineates the individual stages of the cell cycle (Fig. 7b, c). We performed single-cell tracking of 35 cells across the micropattern (20 at the centre and 15 at the edge) over a 45-h period following LIF withdrawal. When assessing the length of the G1 phase at different time intervals post-LIF withdrawal, intriguing differences emerge between cells at the micropattern's edge and those at the centre. Specifically, cells at the micropattern's centre that enter the

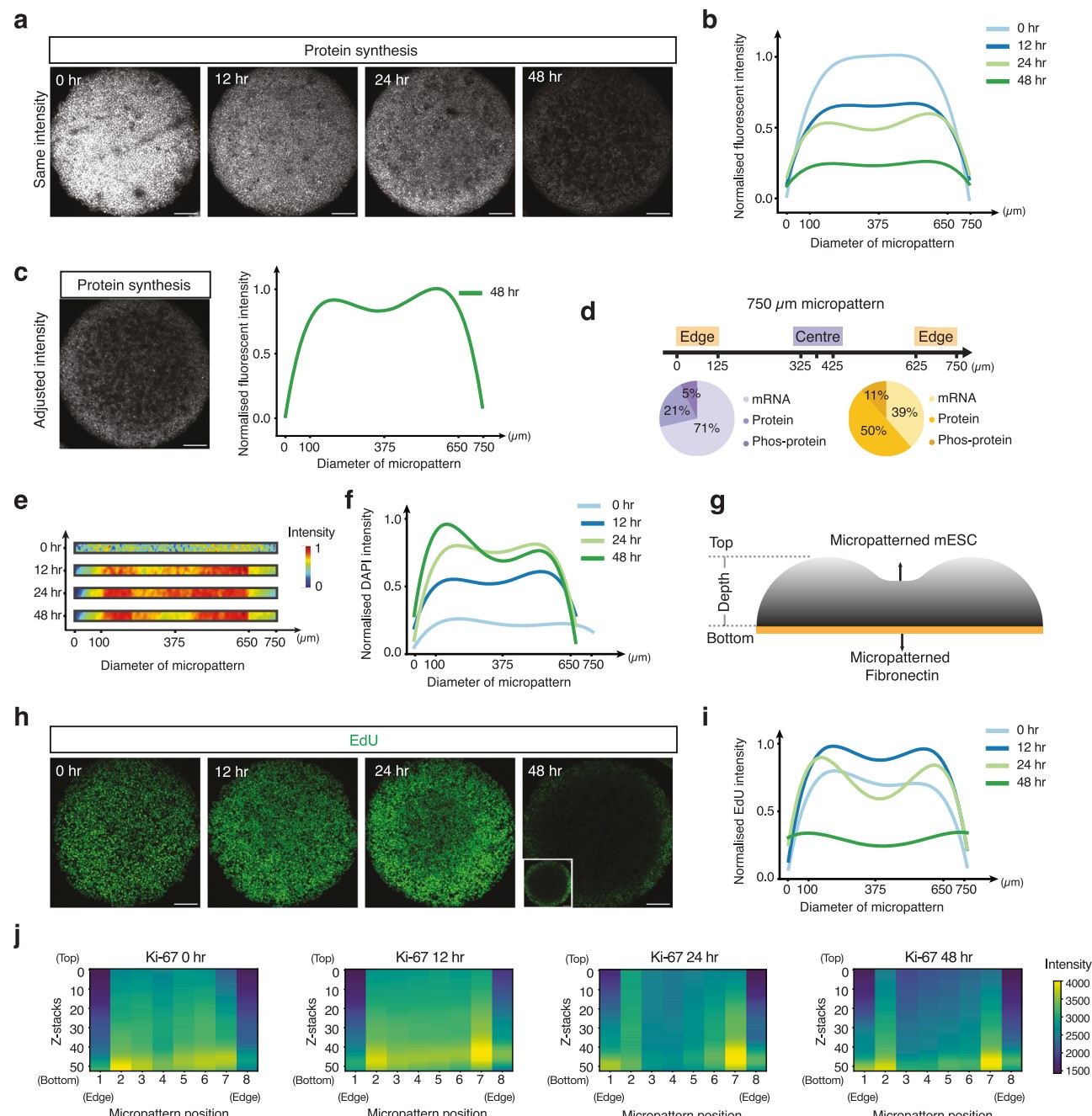

**Fig. 6 | mESCs display position-dependent differences in protein synthesis and proliferation rates. a** Representative images showing the protein synthesis of the micropatterned mESCs at 0, 12, 24 and 48 h after LIF withdrawal. Images from different time points are set as the same maximum and minimum grey value Scale bar, 120 µm. **b** Quantification of relative fluorescent intensity of the composite image of 2–9 micropatterns of the protein synthesis in mESCs at 0, 12, 24 and 48 h after LIF withdrawal (individual intensity profiles of micropatterns in Source Data). **c** Representative images showing the protein synthesis of the micropatterned mESCs at 48 h after LIF withdrawal (left). Scale bar, 120 µm. Quantification of relative fluorescent intensity of the protein synthesis in mESCs across 10 micropatterns at 48 h after LIF withdrawal (right). **d** Pie chart showing the percentage of mRNAs, proteins, and phosphoproteins of targets that are more abundant at the centre (purple) and targets that are more abundant at the edge (orange) of the micropattern. **e, f** Heatmap of DAPI intensity across the micropattern and normalised DAPI intensity across the micropattern at 0, 12, 24 and 48 h after LIF withdrawal. **g** Schematic shows the 'donut' shape of the micropatterned mESCs. **h** Representative images showing EdU staining of the micropatterned mESCs at 0, 12, 24 and 48 h after LIF withdrawal. Images from different time points are set as the same maximum and minimum grey value. Insets: an image of EdU staining at 48 h after LIF withdrawal with adjusted contrast. Scale bar, 120 µm. **i** Quantification of relative fluorescent intensity of EdU staining in mESCs of composite image of 3–10 micropatterns (individual intensity profiles of micropatterns in Source Data) at 0, 12, 24 and 48 h after LIF withdrawal. **j,** Heatmap of Ki-67 expression level at z-projection across 1 micropattern at 0, 12, 24 and 48 h after LIF withdrawal. Source data are provided as Source Data sheet tabs Fig. 6a–j.

G1 phase within the initial 5 h of LIF withdrawal consistently exhibit a relatively short G1 phase of ~3 h (Fig. 7d, top and Supplementary Fig. 31a, left). Conversely, cells that enter the G1 phase after this initial 5-h period show a notably extended G1 phase (Fig. 7e). Markedly, cells at the micropattern's edge display a G1 phase duration similar to cells at the centre (Fig. 7d, bottom and Supplementary Fig. 31a, middle). However, a subset (about 25%) of cells at the edge already exhibit a prolonged G1 phase within the first 5 h of LIF withdrawal

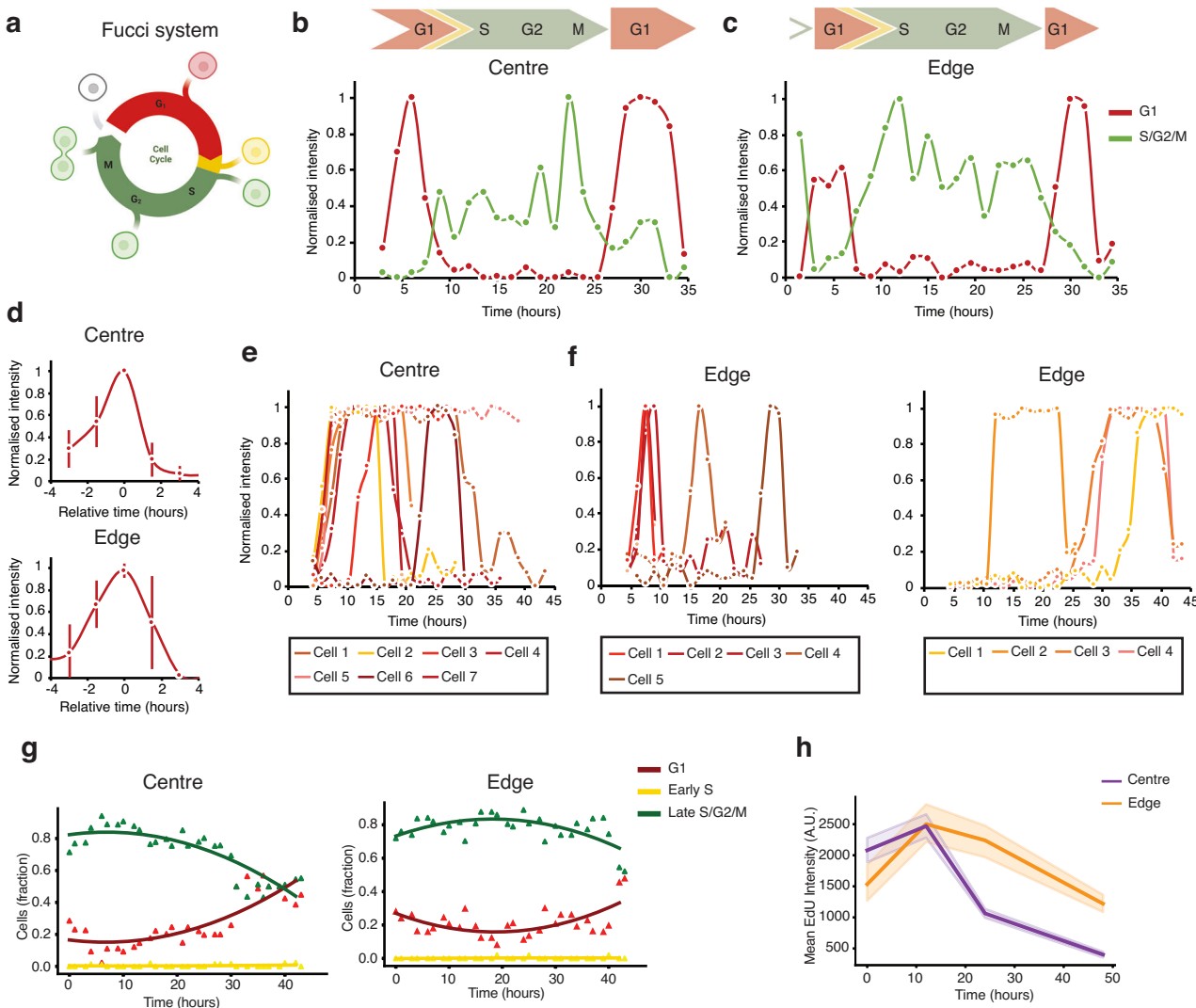

**Fig. 7 | Live cell imaging reveals differences in cell-cycle progression.**
**a** Schematic illustration of the Fucci mESCs demonstrating different colours at different cell cycle stages. Created with BioRender.com. **b**, **c** Kusabira Orange-2 (KO-2) (G1) and Azami Green (Az1) (S/G2/M) intensity of a tracked cell in the centre (**b**) and edge (**c**) of the micropattern. **d** Average KO-2 intensity throughout the G1 phase for cells at the centre (top) and edge (bottom) of the micropattern that enters G1 within 5 h of LIF withdrawal. The average is 12 cells for the centre and 6 cells for the edge. The line represents the mean. Error bars represent standard deviation. **e**, **f** Single cell KO-2 (G1) traces of cells that enter G1 after 5 h of LIF withdrawal at the centre (**e**) and edge (**f**) of the micropattern. 7 (centre) and 9 (edge) cells were analysed respectively. **g** Quantification of the fraction of G1, early S, and late S/G2/M mESCs at the centre (left) and the edge (right) of the micropattern over time after 45 h of LIF withdrawal. 1961 cells at the edge and 2190 cells at the centre were analysed in total. **h** Quantification of mean EdU intensity of mESCs for ten different positions at the edge and the centre of 1 micropattern at 0, 12, 24 and 48 h after LIF withdrawal. The line represents the mean. Shaded regions represent 95% confidence intervals. Source data are provided as Source Data sheet tabs Fig. 7b–j.

(Supplementary Fig. 31a, right). Similarly, after the initial 5-h period, just over half (~60%) of cells closer to the edge continue to cycle, while the remaining portion (about 40%) display an extended G1 phase (Fig. 7f). These single-cell analyses show that the G1 phase of cells at the centre gradually elongates, while >half of the cells at the edge continue cycling until ~30 h post-LIF withdrawal. These data suggested that the cell cycle pace is more heterogeneous at the edge of the micropattern than at the centre.

We next performed a more comprehensive population-based analysis spanning the entire micropattern over the course of the 45-h LIF withdrawal. By comparing the proportion of cells in G1 versus S/G2/M phases for cells at the micropattern's edge and centre over time, we again observe a subtle yet discernible difference (Fig. 7g). Specifically, in the centre, there was a gradual decline in the fraction of cells in S/G2/M phase, beginning ~6 h post-LIF withdrawal (Fig. 7g, left). Instead, cells at the edge of the micropattern show a relatively unchanged

distribution across the cell cycle stages over time, with the proportion of cells in the S/G2/M phase only declining after ~40 h of LIF withdrawal (Fig. 7g. right). Since these data align with the single-cell analysis, the latter is likely representative of the full population. Furthermore, these findings are consistent with the Ki-67 staining results (Fig. 6j and Supplementary Fig. 27), and the cell cycle stages quantified from DAPI intensity differences[66–68] (Supplementary Figs. 29, 30). Although differences in the timing of these changes might arise because the *Fucci* mESCs are a separate cell line[64,65], the fraction of cells in the late S/G2/M phase (*Fucci* mESCs) follow a similar trend to the EdU staining performed on mESCs (Fig. 7h). Lastly, single-cell tracking of both NANOG-GFP and *Fucci* mESCs suggests that cell migration is minimal (Supplementary Fig. 31b, c, respectively). While the potential for rapid cell migration that eludes cell tracking exists, the prevailing data indicates that cells remain either at the centre or the edge of the micropattern (Supplementary Fig. 31c). Collectively, the data indicate that cells

closer to the micropattern's edge exhibit increased proliferation compared to those at the centre.

## Discussion

Thus far, there have been two major areas of focus when studying micropatterned cells: (i) implementation of immunohistochemistry (IHC) to determine spatial-temporal protein abundance in low throughput[10–15]; or (ii) revealing the emergence of specific cell types by single-cell transcriptomic sequencing in high throughput[15]. Here, we developed and exploited a technique that provides quantitative spatial information on different functional proteins and their respective mRNAs at the same time.

ARTseq-FISH, enables simultaneous measurement and quantification of relative mRNA and (phospho-)protein abundance in individual cells. The combined protocol of mRNA and protein detection provides the same resolution of different molecular species and is scalable (Fig. 1). Additionally, the custom image analysis pipeline in ARTseq-FISH (Fig. 2) identifies the cellular nucleus and FISH signals, which allowed detection of 67 mRNAs, proteins, and phosphoproteins simultaneously (Fig. 3). We applied ARTseq-FISH to identify position-dependent gene expression profiles on micropatterned mESCs (Fig. 4). The analysis revealed the expression levels of individual cells and their dependence on the respective location on the micropattern during 48 h of LIF withdrawal. Intriguingly, cells located at the edge and centre of the micropattern obtain different gene expression profiles over time (Fig. 5) while maintaining a high degree of cell-to-cell variability in expression levels. We demonstrate that depending on the micropattern location, cells display differences in protein synthesis, cell cycling and proliferation (Figs. 6, 7). Taken together, our findings indicate that while gene expression is still highly heterogeneous, differences in mRNA and protein levels emerge as early as 12 h after LIF withdrawal.

While other impressive spatial-omics methods that simultaneously measure mRNA and proteins exist (DSP[18] and SMI[36], introduced by NanoString; π-FISH[69]), we focused on developing a protocol that allows for a combined amplification step yielding comparable resolution for mRNA and protein detection. This combined detection and analysis for mRNA and protein offers the opportunity to circumvent the challenges in the integration of different datasets[70–72]. Since the sensitivity of ARTseq-FISH relies on the antibodies (for proteins) and PLPs (for mRNAs), similar limitations apply compared to existing methods[33,34,41,43]. Although there is a risk that the affinity of the antibodies may be influenced by the conjugation method, it is a widely used procedure in current proteomic technologies[33–35]. While we have shown that the detection efficiency of the PLP-based method depends on the binding efficiency of the chosen padlock probe sequence, we only used one PLP to target each mRNA in this study. This is feasible, when comparing relative RNA abundance, either between individual cells (similar to scRNAseq) or, as we do here, between different locations or different time points during the initial stages of mESCs differentiation.

In future, ARTseq-FISH has the potential to detect thousands of targets simultaneously by introducing more unique readout probe sequences and maximising the hybridisation rounds. For instance, to detect 2048 targets, we will need 40 different readout probe sequences and at least ten rounds of hybridisation. However, in theory, sequential FISH methods are able to perform 81 hybridisation rounds, which would allow the detection of more than 10,000 targets with ARTseq-FISH. Yet, to achieve this, it will be critical to assign different hybridisation rounds to high abundance targets to avoid optical crowding. A future application of ARTseq-FISH lies specifically in capturing the spatial information of different molecular species (i.e., mRNAs, proteins, phosphoproteins) and quantifying their relative abundance in particular biological processes. One of the advantages of using padlock probes is the power of signal amplification by RCA.

Therefore, ARTseq-FISH has the potential to detect DNA (i.e., single nucleotide polymorphisms (SNPs))[73], miRNAs[74,75], viral RNA[76] and proximity of proteins[37,77] or even proximity of RNA and protein. In addition, due to the signal amplification of RCA, the results of ARTseq-FISH can also be read out by flow cytometry[78,79], which makes this and future techniques more broadly accessible.

## Methods

### Cell culture

mESC-E14 (mESCs) in this study were obtained from Dr. Hendrik Marks' group at Radboud University[80]. The culture plates were coated with 0.1% gelatin (Sigma-Aldrich, 48723-500 G). Undifferentiated mESCs were cultured and maintained in the medium consisting of high glucose-Dulbecco's Modified Eagles Medium (Gibco™, 41965039) supplemented with 15% foetal bovine serum (Gibco™, A3840002), 2 mM L-glutamine (Gibco™, A2916801), 1 mM sodium pyruvate (Gibco™, 11360039), 1% penicillin–streptomycin (Gibco™, 15140122), 0.1 mM beta-mercaptoethanol (Gibco™ 31350010) and 500 U/mL recombinant leukaemia inhibitory factor (LIF; Millipore, ESG1107) at 37 °C in 5% CO₂. Cells were passaged every 2–3 days using 0.05% Trypsin/EDTA (Gibco™, 25300062). Cell density was $5 \times 10^4$ per well for the experiments in the μ-Slide (ibidi, 81501) and $1 \times 10^4$/cm² for the experiments in the μ-Slide 3D perfusion (ibidi, 80376). For differentiation assays, cells were cultured on micropatterns in LIF medium for 1 day and then changed to a medium without LIF (-LIF) for up to 2 days.

### Micropatterned substrate fabrication

Micropatterned islands with 750 μm diameter were produced on a silicon mask using standard photolithography[81]. Distances between the microfeatures were 100 μm and the height of the microfeatures was between 50 and 100 μm. The PDMS and the curing agent (SYLGARD 184, 1.1KG) was mixed with a ratio of 10:1 in a plastic beaker. The PDMS mixture was degassed to remove the air bubbles under vacuum. Then, the degassed PDMS mixture was poured onto the master in a petri dish. A gel was formed in a 60 °C oven overnight. Finally, the gel PDMS stamps was peeled off the master.

The μ-Slide 3D perfusion microchambers were plasma oxidised to improve protein printing efficiency and cell attachment on the glass bottoms[82]. The surface of the PDMS stamps was placed with a drop of 50 μg/mL fibronectin in PBS and incubated for 1 h at room temperature. The PDMS stamps were washed twice with PBS and dried thoroughly. The glass bottoms of μ-Slide 3D perfusion were then coated with the fibronectin from the PDMS stamps, followed by the graft of the PLL (20)-g [3.5]- PEG (2) (0.1 mg/mL, SuSoS AG) in 10 mM HEPES (Gibco™, 15630080) for 30 min at room temperature. The PLL-g-PEG prevents cell attachment in the nonprinted areas[83].

### Immunostaining

Cells were fixed with 4% paraformaldehyde (PFA) (Sigma-Aldrich, 1040021000-1 L) for 10 min at room temperature and then permeabilized with 0.5% Triton in 1x PBS at room temperature for 10 min. Cells were washed with 1x PBS three times after fixation and permeabilization. After permeabilization, cells were incubated in a blocking buffer (5% BSA in 1x PBS containing 0.5% Triton) for 1 h at room temperature. The primary antibodies were diluted in a blocking buffer and incubated at room temperature for 2–4 h. Cells were then washed with 1x PBS three times and incubated in a secondary antibody solution for 2 h at room temperature. The secondary antibodies were diluted 1:1000 in 1x PBS. DAPI (10 μg/mL) (Thermo Scientific™ 62247) staining was performed together with secondary antibodies.

For Ki-67 (Cell Signaling, 9192), Caspase-3 (Cell Signaling, 94530), OCT4 staining, the primary antibodies were diluted as recommended ratio in blocking buffer. The cells were then incubated overnight at 4 °C in the primary antibody solution. The other steps of the protocol remained the same as above.

## smFISH

Probes of *Nanog* were purchased from Stellaris (LGC Biosearch Technologies, Novato, CA) (http://www.singlemoleculefish.com/). Each set contains 28 probes and each probe is 20 nt long (Supplementary Data 1). Probes are conjugated with TAMRA. Cells were fixed with 4% PFA (Sigma-Aldrich, 1040021000-1 L) for 10 min at room temperature and then permeabilized with pre-chilled 70% EtOH at 4 °C for a minimum of 1 h. Cells were washed three times with 1x PBS. Probes were diluted to a final concentration of 50 nM in 1 g/mL dextran sulfate (Sigma, catalogue # 42867), 2x SSC and 10% formamide (Invitrogen, AM9342). The hybridisation of the probes was performed overnight at 37 °C. The next day, samples were washed with 2x SSC buffer at 37 °C for 30 min and then incubated with DAPI solution (diluted in 2x SSC) at 37 °C for 15–20 min. Cells were washed with 2x SSC and then switched to a photo-protective buffer containing 50% glycerol (Thermo Fisher Scientific, Waltham, MA), 75 µg/mL glucose oxidase (Sigma-Aldrich, Darmstadt, Germany), 520 µg/mL catalase (Sigma-Aldrich, Darmstadt, Germany) and 0.5 mg/mL Trolox (Sigma-Aldrich, Darmstadt, Germany). Samples were imaged with a spinning disk confocal microscope (OLYMPUS PlanXApo 60x/1.40- N.A. oil-immersion objective, and Andor iXon camera). DAPI was excited by 405 nm, and TAMRA was excited by 540 nm lasers. The distance of Z-stacks was set to 0.5 µm. The exposure time and laser intensity were set to reach a maximum grey value of 16,000 for each image.

## Probe design of ARTseq-FISH

The mRNA sequences were obtained from the NCBI website (*Mus musculus*). Probes were designed according to the targeted regions by using online software (http://biosearchtech.com/stellaris-designer/), for individual probe sequences see Supplementary Data 1. For targeting RNA directly, we selected 36 nt from the mRNA sequence (18 nt for 5′ arm and 18 nt for 3′ arm of PLP). Each PLP contains two arms (18 nt each) for binding to the targeting region, one unique 28 nt sequence identical to the bridge probe and one common RCA primer binding region. Besides the 28 nt sequences identical to paired PLPs, BrPs consist of four overhang sequences with 18 nt each.

Oligonucleotides used for antibody conjugation consist of a 10 nt linker (poly-A) and a 36 nt sequences corresponding to the complementary PLP. The 36 nt sequences for antibody conjugation, 28 nt sequences of PLPs and BrPs, and 18 nt readout probe sequences were identified using a previously published algorithm[24]. The readout probes are 18 nt in length and labelled with three different fluorophores (ATTO 488, TAMRA, CY5). We used BLAST from NCBI to check that the 18 nt probe sequence is less than 13 nt (<70%) complementary to other Mus musculus transcripts. All the probes have GC content within the range 45–65% and were purchased from Integrated DNA technologies (IDT).

| Oligonucleotides (46 nt) | 5′ - Azide-AAAAAAAAAA- [36 nt unique sequence] −3′ |
|---|---|
| Padlock probe (84 nt) | 5′ -phosphorylated- [18 nt 5′-arm] - [28 nt unique sequence] - [20 nt RCA primer] - [18 nt 3′ - arm] − 3′ |
| Bridge probe (108 nt) | 5′ - [readout 1] -AA- [readout 2] -AA- [28 nt unique sequence] -AA- [readout 3] -AA- [readout 4] − 3′ |

## Antibody−DNA conjugation

Antibodies (Abs) used in ARTseq-FISH were labelled with DNA oligonucleotides as previously described[39]. In brief, Abs were first buffer exchanged into 0.2 M NaHCO3 (pH 8.3) (Thermo Scientific Across 217125000) by using Zeba™ Spin Desalting Columns, 40 K MWCO (Thermo Scientific™ 87767). Abs were then reacted with 10x molar excess of dibenzocyclooctyne-*S*-*S*-*N*-hydroxysuccinimidyl (DBCO-S-S-NH) ester (Sigma-Aldrich, 761532-1 mg) for 2 h at room temperature in the dark. Next, Abs were purified using Zeba™ Spin Desalting Columns, 40 K MWCO, to remove excess DBCO-s-s-NHS. 5′ Azide modified DNA oligonucleotides (IDT) were incubated with Abs (molar ratio of antibody to single-stranded DNA of 1:15) overnight at 4 °C in the dark. The conjugated Abs were buffer exchanged into PBS (Gibco™ 20012019) containing 0.05% sodium azide (Sigma-Aldrich, S8032-25G) and 0.1 mM EDTA (Lonza, 51201), and excess oligos were removed by 100 K Amicon centrifuge filters (Merck UFC510096). The DNA-conjugated Abs were stored at 4 °C. The list of antibodies and corresponding DNA oligonucleotides is provided in Supplementary Data 1 and Supplementary Data 2. The dilution of conjugated antibodies was optimised between ratios of 1:1000, 1:2000 and at 1:4000 and the ratio of 1:4000 was used for ARTseq-FISH.

## ARTseq-FISH

Fixation and permeabilization: Cells were fixed with 4% PFA (Sigma-Aldrich, 1040021000-1 L) for 10 min at room temperature and then permeabilized with pre-chilled 100% methanol at −20 °C for 10 min. Cells were washed with 1x PBS three times after fixation and permeabilization.

Blocking and Primary antibody incubation: After permeabilization, cells were incubated in a blocking buffer (5% BSA in PBS containing 0.5% Triton) for 1 h at room temperature. The DNA oligonucleotide labelled primary antibodies were diluted to 1:4000 in blocking buffer and incubated at room temperature for 2–4 h.

RNA PLP hybridisation and ligation: Phosphorylated PLPs were hybridised to the mRNA sequences directly at a final concentration of 100 nM/PLP. The reaction was performed overnight at 37 °C using 1x Ampligase ligase buffer (Lucigen, A1905B), 50 mM KCl (Invitrogen, AM9640G), 20% Formamide (Invitrogen, AM9342), 0.2 µg/µL tRNA, 1 U/µL RiboLock RNase Inhibitor (Thermo Scientific, EO0381) and 0.2 µg/µL of BSA in Nuclease-free water (Invitrogen, AM9937). Next, PLP ligation was performed for 4 h at room temperature using 0.5 U/µl SplintR Ligase (NEB, M0375), 1x Ampligase ligase buffer, 10 µM ATP (Invitrogen, 18330019) and 0.2 µg/µL of BSA in Nuclease-free water.

Ab PLP hybridisation and ligation: The hybridisation of PLPs to DNA oligonucleotides conjugated to Abs was performed at a final concentration of 5 nM/PLP. The ligation of PLPs was performed in the same reaction with T4 DNA ligase, 1x T4 DNA ligase buffer, 50 mM KCl (Invitrogen, AM9640G), 20% formamide (Invitrogen, AM9342), 0.2 µg/µL of BSA and Nuclease-free water. This reaction was incubated at 37 °C for 30 min followed by 45 °C for 90 min.

Rolling circle amplification (RCA): RCA was performed with 0.25 U/µL Phi29 polymerase (NEB, M0269L), 1x Phi29 polymerase buffer, 5% Glycerol (Sigma, G5516), 0.25 mM each dNTP (Thermo Scientific, R0192), 0.2 U/µL Exonuclease I (Thermo Scientific, EN0581), 0.2 µM RCA primer and 0.2 µg/µL of BSA in Nuclease-free water overnight at 30 ˚C.

Bridge probe hybridisation: BrPs were hybridised at a final concentration of 100 nM per probe. The reaction took place in the hybridisation buffer, which contains 2x SSC, 20% formamide, 0.05 g/mL dextran sulfate and nuclease-free water, for 1–2 h at room temperature. An extra fixation with 4% PFA was performed to stabilise the binding of bridge probes and rolling circle amplification products.

Readout probe hybridisation: RoPs were hybridised at a final concentration of 100 nM per probe and in the same hybridisation buffer as the BrPs. The reaction can be performed together with DAPI (10 µg/mL) (Thermo Scientific™ 62247) staining for 1 h at room temperature. Cells were washed with 2x SSC and then switched to a photo-protective buffer containing 50% glycerol (Thermo Fisher Scientific, Waltham, MA), 75 µg/mL glucose oxidase (Sigma-Aldrich, Darmstadt, Germany), 520 µg/mL catalase (Sigma-Aldrich, Darmstadt, Germany) and 0.5 mg/mL Trolox (Sigma-Aldrich, Darmstadt, Germany).

Image acquisition: Images of sequential hybridisation were acquired with a spinning disk confocal microscope (OLYMPUS Plan-XApo 60x/1.40- N.A. oil-immersion objective, and Andor iXon camera). DAPI was excited by 405 nm, and ATTO 488, TAMRA, CY5 was excited by 475, 540 and 638 nm lasers respectively. The distance of Z-stacks was set as 0.5 μm. The exposure time and laser intensity were set differently in different channels to obtain the same maximum intensity value of spots.

Stripping: Samples were incubated in 60% formamide solution (60% formamide in 2x SSC) for 10 min to remove the readout probes after image acquisition and then washed three times with 2x SSC.

Rehybridisation: A new set of RoPs were hybridised in the hybridisation buffer for 1 h at room temperature in dark.

Image acquisition: Same as described above. Repeat the cycle of image acquisition, stripping off readout probes and rehybridisation.

### Single-cell RNA sequencing (scRNAseq) in mESCs

mESC were cultured as previously described. Cells were grown for 30 h in a 37 °C incubator with 5% $CO_2$. Cells were exposed to 0.007% EtOH for the last 6 h. Cells were then washed once with PBS and detached by 0.05% trypsin-EDTA for 2 min at 37 °C. Detached cells were pelleted and resuspended in 1 mL of freezing media consisting of 80% culture media, 10% of extra heat-inactivated FBS and 10% of DMSO. The proper viability of the cells prior to freezing was quantified with propidium iodide/acridine orange staining in a LUNA-FL Dual Fluorescence Cell Counter. Frozen cells were delivered to the commercial company Single Cell Discoveries (https://www.scdiscoveries.com) in dry ice. Single-cell barcoding was performed in a 10x microfluidic genomic chip in order to encapsulate individual cells in water droplets in oil, containing cell-specific barcoded beads. Labelled RNA molecules were pooled and subjected to a poly-A-specific reverse transcription. cDNA molecules where linearly amplified using an in vitro transcription reaction, and the final sequencing library was obtained through a second step of reverse transcription. 3′-end sequencing was performed, followed by quality control and genomic mapping. Final read count per gene and per cell matrices were generated and used in posterior analysis steps.

### Flow cytometry analysis of cell cycle

mESCs were seeded in 1.9 $cm^2$ wells (24-well format) with a density of $4.4 \times 10^3$ cells/$cm^2$. The wells were previously coated with 0.1% gelatin. Cells were seeded in serum/LIF medium and cultured for 24 h. Then, cells were fixed in pre-chilled 70% ethanol for 30 min at −20 °C and washed with 1x PBS. Finally, cells were stained with 1 μg/mL propidium iodide (Sigma-Aldrich, P4864-10ML) containing 10 μg/mL RNaseA (VWR International B.V, 0675-250MG) for 15 min. After staining, cells were transferred to a flow cytometry tube and kept on ice until their immediate analysis in a BD FACS Calibur™ flow cytometer.

### Flow cytometry analysis of NANOG-GFP mESC cell line in serum/LIF conditions

The NANOG-GFP mESC line was kindly provided by Prof. Dr. Leor Weinberger's group (Gladstone Institutes)[57]. NANOG-GFP mESCs and non-labelled mESCs were seeded in 9.5 $cm^2$ wells (six-well format) with a density of $4 \times 10^4$ cells/$cm^2$. The wells were previously coated with 0.1% gelatin. Cells were seeded in serum/LIF medium. Twenty-four hours post-seeding, cells were detached with 400 μL of 0.05% trypsin-EDTA for 2 min and resuspended in growing media. Cells were transferred to a flow cytometry tube and kept on ice until their immediate analysis in a BD FACS Calibur™ flow cytometer.

### Flow cytometry analysis of CD24 expression in mESC

mESCs were seeded in 1.9 $cm^2$ wells (24-well format) with a density of $4.4 \times 10^3$ cells/$cm^2$. The wells were previously coated with 0.1% gelatin. Cells were seeded in serum/LIF medium and cultured for 96 hours,

with a medium renewal at 48 h of culture. Cells were stained with an anti-mouse CD24 antibody conjugated to alexa647 fluorophore (BioLegend, 101818) or an alexa647 isotype control antibody (BioLegend, 400626) for 30 min in serum/LIF media at 37 ˚C. After staining, cells were washed with 1x PBS and detached with accutase (StemCell™ technologies, 07922) for 5 min at 37 ˚C, transferred to a flow cytometry tube and kept on ice until their analysis in a BD FACSVerse™ flow cytometer.

### EdU proliferation assay

mESCs were seeded onto the micropatterns, ensuring complete coverage, and cultured for 16 h. Subsequently, the medium was changed to a LIF-free medium. To assess cell proliferation at different time points, the culture medium was supplemented with 10 μM of EdU two hours before fixation at each time point. The detection of EdU incorporation was performed according to the protocol provided by the EdU proliferation kit (Abcam, ab219801, iFluor 488). In brief, the cells were washed twice with Wash Buffer and then fixed with 4% PFA for 10 min at room temperature, while avoiding exposure to light. After the fixation step, the cells were washed twice and permeabilized with 1x Permeabilization Buffer for 20 min at room temperature. Subsequently, the cells were washed twice and incubated in the dark for 30 min at room temperature in a Reaction mix containing TBS, 4 mM Copper Sulfate, 1.2 μM iFluor 488 azide dye (500 μM in DMSO), and 1x EdU additive solution. Following another round of washing, the cells were stained with Hoechst 33342 (5 μg/mL). Finally, the cells were imaged using a spinning disk confocal microscope (OLYMPUS UPlanFL N 10x/0.30 N.A. objective and Andor iXon camera). The Hoechst 33342 and EdU signals were excited at wavelengths of 405 and 475 nm, respectively.

### Protein synthesis assay

mESCs were seeded onto the micropatterns, ensuring complete coverage, and cultured for 16 h. Subsequently, the medium was changed to a LIF-free medium. To assess the protein synthesis at different time points, the culture medium was supplemented with 20 μM of Click-iT® OPP 30 min before fixation at each time point. The detection of protein synthesis was performed according to the protocol provided by the Click-iT® Plus OPP Protein Synthesis Assay Kits (Thermo Fisher, C10456). In brief, the cells were washed once with 1x PBS and then fixed with 4% PFA for 10 min at room temperature. After the fixation step, the cells were washed twice and permeabilized with 0.5% Triton-X-100 in PBS for 15 min at room temperature. Subsequently, the cells were washed twice and incubated in the dark for 30 min at room temperature in a Click-iT® Plus OPP reaction cocktail containing 1x Click-iT® OPP Reaction Buffer, Copper Protectant (Component D), Alexa Fluor® picolyl azide (Component B) and 1x Click-iT® Reaction Buffer Additive. Following another round of washing, the cells were stained with 1X HCS NuclearMask™ Blue Stain for 30 min at room temperature, protected from light. Finally, the cells were imaged using a spinning disk confocal microscope (OLYMPUS UPlanFL N 10x/0.30 N.A. objective and Andor iXon camera). The NuclearMask™ Blue Stain and protein synthesis signals were excited at wavelengths of 405 and 638 nm, respectively.

### Apoptosis detection

To detect apoptosis of mESCs, Cleaved Caspase-3 (Cell Signaling, 94530) and 7-AAD (Thermo Fisher, 00-6993-50) were measured to assess the middle and late stage of apoptosis respectively. mESCs were seeded onto the micropatterns, ensuring complete coverage, and cultured for 16 h. Subsequently, the medium was changed to a LIF-free medium. To assess the middle stage of apoptosis after 48 h differentiation, fixed cells were stained with Caspase-3 (cleaved) as mentioned in the section of " Immunostaining". To assess the late stage of apoptosis after 48 h differentiation, live cells were stained with 7-AAD

and Hoechst 33342 (Sigma-Aldrich, B2261-25MG) for 30 min at a 37 °C incubator. Finally, the cells were imaged using a spinning disk confocal microscope (OLYMPUS UPlanFL N 10x/0.30 N.A. objective and Andor iXon camera). Quantification of apoptosis from nuclear morphology was performed on DAPI images acquired by ARTseqFISH as previously described[62].

## Live cell imaging

Fucci mESCs[64,65] were kindly provided by Menno Ter Huurne and Hendrik Marks at Radboud University. NANOG-GFP mESCs[57,58] and Fucci mESCs were seeded onto the micropatterns, ensuring complete coverage, and cultured for 16 h. Subsequently, the medium was changed to a LIF-free medium, and live cell imaging was performed after LIF withdrawal. Imaging was conducted using the SP8x AOBS-WLL confocal laser scanning microscope, equipped with a Leica 40x/ 1.1 Water objective and a live-cell environmental chamber maintained at a humidified temperature of 37 °C and 5% $CO_2$. For NANOG-GFP mESCs, images were acquired using a HyDs detector and a 488 nm laser. As for Fucci mESCs, images were acquired using a HyDs detector, along with both 488 and 561 nm lasers. Z-series was obtained with a step size of 1.5 µm. The time interval between image acquisitions was set to 1.5 h. Single-cell tracking was performed manually using the Multi-point tool in Fiji[84].

## Image and data analysis

Details described in Supplementary Information, Supplementary Note 2: Computational pipeline development of ARTseq-FISH. Parts of the software developed for ARTseq-FISH have been applied in our previously published work[85–87].

## Statistics and reproducibility

Figures 1b, 2b, 4b, 6a, c, h are validation experiments. For Fig. 1b, a single experiment was performed for each target. For Fig. 2b, we chose one position of one micropattern presented in the main text datasets. For Fig. 4b, the immunostaining results, we selected three antibodies (BRACHYURY, POL II and p53) because ARTseq-FISH data indicated they were either highly expressed at the edge, in the centre or equally present in both. For Fig. 3e, for TAMRA and CY5, the intensity distributions between the detected signal and the signal of hybridised probes did not overlap (Source Data tab Fig. 3e), thus these did not present a potential misassignment risk. For the ATTO 488 channel, there was an overlap between distributions. The images of three negative controls and one ARTseq-FISH experiment were merged separately, the negative control with the smallest reduction in false positives is shown (Source Data tab Fig. 3e). For Fig. 6a, a single experiment was performed, and between 2–9 micropatterns were imaged between time points 0 and 48 h, the intensity profiles of the individual micropatterns are given in Source Data tab Fig. 6a. Figure 6b was a composite image of all micropatterns per time point (Source Data tab Fig. 6b). For Fig. 6h a single experiment was performed and between 3–10 micropatterns were imaged between time points 0 and 48 h. Similar to Fig. 6a, b, the individual intensity profiles of the micropatterns and the profile of the composite image per time point are given in Source Data Fig. 6h, i, respectively. For the time-lapse experiment in Fig. 7, a single experiment and micropattern was imaged (Source Data tab Fig. 7).

## Reporting summary

Further information on research design is available in the Nature Portfolio Reporting Summary linked to this article.

## Data availability

All relevant data supporting the key findings of this study are available within the article and its Supplementary Information files. Sample images to test software are provided together with software, see archive DOI: 10.5281/zenodo.10723692 and github.com/Hansen-Labs/ ARTseqFISH. Raw imaging data is stored on a local drive. The data used to train the SVM models, which detects nuclei and the point spread functions of hybridised signals, can also be found in the archive. Sequencing data associated with this article has been deposited in NCBI GEO under accession number GSE264348. Source data are provided with this paper.

## Code availability

The package is written in Python 3.8 (python software foundation, Delaware US). Code can be found at Hansen-Labs GitHub at http:// github.com/Hansen-Labs/ARTseqFISH, code archived, see archive DOI: 10.5281/zenodo.10723692 or github.com/Hansen-Labs/ARTseqFISH. For more information about software contact bob.vansluijs@gmail.com.

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

## Acknowledgements

We thank members of the Huck and Hansen laboratories for the thoughtful discussion and suggestions; Dr. Aigars Piruska for making the silicon masters with micropatterns and technical assistance with CSU spinning disk confocal; Dr. Erik van Buijtenen and Dr. Kinga Matuła for sharing the antibody conjugation protocols, and fruitful discussions and suggestions from Dr. Erik van Buijtenen. We also thank Prof. Dr. Leor Weinberger for the NANOG-GFP mESC line as well as Dr. Hendrik Marks for the mESC-E14 and mESC Fucci cell line and thoughtful suggestions. We acknowledge generous support from Radboud University (W.T.S.H.), the Christine Mohrmann fellowship (M.M.K.H.), the European Union (ERC, ChOICE, 101041939, M.M.K.H.), the Netherlands Organisation for Scientific Research (NWO) (ENW-XS awards OCENW.XS3.055 and OCENW.XS21.2.050, both M.M.K.H., and a Spinoza grant, W.T.S.H.), and Oncode Institute, which is partly financed by the Dutch Cancer Society (M.M.K.H.). X.H. acknowledges financial support from the China Scholarship Council (CSC).

## Author contributions

X.H., B.v.S., W.T.S.H. and M.M.K.H. conceived and designed the study. X.H. developed and optimised the experimental procedure for ARTseq-FISH. B.v.S. developed the software for ARTseq-FISH. Ó.G.-B. helped with probe design and optimisation of ARTseq-FISH workflow. Ó.G.-B. also conducted supporting experiments. Y.S. and K.R. contributed to the validation experiments of ARTseq-FISH. X.H., B.v.S., W.T.S.H. and M.M.K.H. wrote the manuscript and Ó.G.-B. edited the manuscript.

## Competing interests

The authors declare no competing interests.
