## [Peer Review File · Nature Communications]

ARTseq-FISH reveals position-dependent differences in gene expression of micropatterned mESCsEditorial Note: This manuscript has been previously reviewed at another journal that is not operating a transparent peer review scheme. This document only contains reviewer comments and rebuttal letters for versions considered at *Nature Communications*.

Reviewer #1 (Remarks to the Author):

The authors have improved the manuscript significantly, but there are still some somewhat "stretched claims" that does not add to the credibility of the work. I think the authors should tune down some of those claims, for example:

The authors mentioned that 'ARTseq-FISH not only allows detection of multiple mRNA and protein targets simultaneously but uniquely provides quantification of low abundance targets--such as RB--due to the signal amplification step we incorporate'. I do not see how signal amplification can help to detect low abundant targets. It is widely accepted that padlock probe and RCA methods are less sensitive as compared to FISH based approaches, thus wouldn't FISH based methods be more suited to detect low abundant targets? Data in Extended Data Fig.6d shows that smFISH is 2,5x more sensitive than ART-seq FISH.

'Therefore, the strength of ARTseq-FISH lies in comparing relative spatial or temporal changes in expression levels of mRNA and proteins' The authors mentioned the supposed 'strength' of the method but do not provide data supporting the claim where one can indeed use ARTseq-FISH to compare spatial or temporal changes in expression levels of mRNA and proteins.

The authors maintain that the RCA spots are diffraction limited (<0,25 micrometer in diameter). Yet, the RCP:s in the insets of extended figures 3 and 4 are of a comparable size as the 1 micrometer scale bar. In their theoretical calculation of the expected size of RCP:s they have introduced an error in the length of the padlock probes. They are not 36 nt long, they are 84 nts long.

In extended data fig 4b, there are quite a number of targets with low spots / cell count that is almost similar to the level of negative control. The validation data presented here does not convince me that the probing approach is sound given that the positive counts for most genes are as low as the negative control.

Minor comments:

Lines 179-180 'Notably, while unphosphorylated RB shows low spot counts at a protein level, its mRNA and phospho-RB display higher spot count' I do not understand the significance / intent of this sentence here, since the authors after commented that one can not compare the absolute counts between species.

Extended Data Fig5 - formatting / spelling error 'Representative max intensity and ...'

Reviewer #3 (Remarks to the Author):

In this manuscript Hu et al develop a new technique known as ARTseqFISH to simultaneously measure protein and mRNA abundances in situ at relatively high throughput (67 targets). They apply this technique to mouse embryonic stem cells (mESCs) cultured on micropatterns and use this to demonstrate that the differentiation is position dependent. They also suggest based on expression of cell cycle markers that the differentiation is caused by cell cycle changes. I am not an expert on spatial transcriptomics or high throughput protein measurements, so I cannot comment on whether the current technique represents an advance over existing methods and will largely restrict my comments to the biological system the authors address. Here, there are significant issues including misunderstandings of this phase of development and a number of conclusions which are not supported by the data. I detail these concerns below.

1. The statement of the authors that "cells continue to self-renew by progressing through the cell cycle, or terminally differentiate by exiting the cell cycle during the G1 phase" is incorrect in this context. No terminal differentiation occurs during the two day time window starting from

pluripotency which the authors study and even cells which have exited pluripotency (e.g. primitive streak cells, mesoderm progenitors, neural progenitors etc) will continue to cycle. This is important because the dichotomy between cycling and differentiation is fundamental to how the authors frame their data but it does not apply in this context.

2. The authors misrepresent what has been shown previously on micropatterns. Particularly the statements "it remains unknown how positioning on the micropattern impacts cellular decisions to ultimately define cell fate[9-15]. Specifically, the sequence of events that influence cell fate decisions on micropatterns and what factors drive the choice between self-renewal and differentiation remain unclear". While there is more to learn, the references the authors cite clearly establish that there is position-dependent differentiation on micropatterns (this is the main reason the system is interesting) and have begun to work out in detail the mechanisms of this differentiation in terms of the signals involved, how they change in time, and how they influence cell fates. Rather than insisting that little is known, the authors need to discuss the new knowledge generated by their study in light of the extensive work that has already been done on micropatterned systems.

3. The discussion of Nodal in lines 213-216 is inaccurate. First, the antibody the authors use for "unphosphorylated" Smad2 (31H15L4) is actually an antibody for total Smad2, both phosphorylated and not. Thus, it should largely agree with mRNA distribution, and that fact that pSmad2 rather than Smad2 agrees with the mRNA distribution is likely the result of experimental error as the differences between edge and center in these factors are small. Moreover, it is not correct that unphosphorylated Smad has "bifunctional roles in the regulation of both self-renewal and differentiation" as Smad performs its primary functions when phosphorylated. It is the phosphorylated form that does this. This example shows the dangers of interpreting the data to fit an expected narrative.

4. There may be a similar issue with the authors discussion of phosphorylation of Rb protein, but I cannot tell as no catalogue or clone number is given for the antibody. All antibodies should have manufacturers and catalogue numbers listed.

5. The discussion of apoptosis and senescence is not supported by the data. p53 expression alone is not sufficient to conclude that apoptosis is occurring, the authors should analyze specific markers of this process. It is unclear what the claims regarding senescence are based on.

6. The discussions of proliferation rate and cell density are also not warranted. First, one cannot make conclusions about proliferation rate from the expression data at alone. The authors need perform live cell imaging to actually determine the interval between cell divisions of individual cells. For example, the statement that "After 48 hours cells at the edge enter and remain in the G1 phase (Fig. 6h), from which they enter the terminal differentiation stage" is not warranted. All that is known from the data is that a somewhat larger fraction of cells are in G1 at 48 hours. This does not necessarily result from terminal differentiation but may reflect a lengthening of the G1 phase of the cycle as cells exit pluripotency. In fact, as noted above, it is not possible for cells to go from pluripotency to terminal differentiation in 48 hours. Second, variations in cell density do not necessarily result from variations in proliferation but can also be due to cell movements. Both in vivo and some micropatterned systems it has been shown that cells migrate into a primitive streak like region and this will result in a local increase in density. Again, live cell imaging is required to distinguish these possibilities.

7. The conclusions about the temporal sequence of events over 48 hours are almost solely inferred from a pseudotime analysis. This is problematic in any case, but is especially problematic for cyclical events such as the cell cycle. In fact, contrary to the claims the authors make in the text, the expression of cell cycle genes in pseudotime is cyclic (fig 6d), not reflecting a steady progression from proliferation to exit from the cell cycle. The fact that the cell cycle genes appear cyclic here raises the possibility that the pseudotime in the authors calculations does not reflect a differentiation trajectory but is simply ordering cells by cell cycle phase. If the pseudotime actually reflects a differentiation trajectory, why do the self-renewal genes begin to increase again at the end?

8. Even if one accepts the authors claims regarding pseudotime, the conclusion that the differentiation events result from cell cycle changes is not warranted. First, some of the differentiation genes such as Snail and Sox17 change as rapidly as the cell cycle genes. Second, even if cell cycle genes do change before differentiation ones, perturbative experiments would be required to establish causation.

9. It isn't clear to me whether the changes in differentiation markers are biologically significant. For example, Bra seems to change about 2 fold over pseudotime but one would expect a much

larger increase in actual primitive streak. The authors should include positive controls (such as established mesoderm differentiation protocols) and quantify the expression of differentiation genes relative to expression in these controls.

10. It seems that cell segmentation is performed using a nuclear image, but many of the targets will be cytoplasmic. The authors do not have a definitive way of associating each spot with the correct nucleus without marking the cell boundaries. This source of error should be acknowledged.

Reviewer comment:

The authors have improved the manuscript significantly, but there are still some somewhat "stretched claims" that does not add to the credibility of the work. I think the authors should tune down some of those claims, for example:

Response:

We thank the reviewer for the overall positive assessment of our revised manuscript. To address the reviewers remaining comments, we have now significantly revised the text as detailed below.

The authors mentioned that ‘ARTseq-FISH not only allows detection of multiple mRNA and protein targets simultaneously but uniquely provides quantification of low abundance targets—such as RB—due to the signal amplification step we incorporate’. I do not see how signal amplification can help to detect low abundant targets. It is widely accepted that padlock probe and RCA methods are less sensitive as compared to FISH based approaches, thus wouldn't FISH based methods be more suited to detect low abundant targets? Data in Extended Data Fig.6d shows that smFISH is 2,5x more sensitive than ART-seq FISH.

Response:

We apologize for the confusion, we meant specifically compared to regular immunofluorescence where there is typically only one fluorophore per antibody. In order to tone down this claim, we have now removed this sentence from the text.

‘Therefore, the strength of ARTseq-FISH lies in comparing relative spatial or temporal changes in expression levels of mRNA and proteins’ The authors mentioned the supposed ‘strength’ of the method but do not provide data supporting the claim where one can indeed use ARTseq-FISH to compare spatial or temporal changes in expression levels of mRNA and proteins.

Response:

With “relative spatial or temporal changes in expression levels” we meant that similar to other omics methods¹⁻³, it is prudent not to use ARTseq-FISH to quantify absolute mRNA and protein counts, but rather to compare relative changes upon addition or removal of a stimulus (i.e., LIF), or between cells located at different positions (i.e., centre or edge of a micropattern). In order to highlight this, we have now amended Fig. 5 to specifically show relative spatial and temporal changes in mRNA and protein abundance. To emphasize this in the text, we have removed the sentence Reviewer 1 mentions and instead solely highlight that absolute values cannot be quantified (see line 186):

“As mentioned previously, similar to other omics methods that rely on antibodies^{33, 34}, ARTseq-FISH is dependent on the binding efficiency of primary antibodies (for proteins) and PLPs (for mRNAs). Therefore, we cannot compare absolute counts between species.”

The authors maintain that the RCA spots are diffraction limited (<0,25 micrometer in diameter). Yet, the RCP:s in the insets of extended figures 3 and 4 are of a comparable size as the 1 micrometer scale bar. In their theoretical calculation of the expected size of RCP:s they have introduced an error in the length of the padlock probes. They are not 36 nt long, they are 84 nts long.

Response:

In the previous draft the argument regarding diffraction limit was removed. In order to further tone down the manuscript we removed “at sub-micrometre spatial resolution” from the abstract.

In extended data fig 4b, there are quite a number of targets with low spots / cell count that is almost similar to the level of negative control. The validation data presented here does not convince me that

the probing approach is sound given that the positive counts for most genes are as low as the negative control.

Response:

We apologize for not having made this clearer. The image analysis pipeline, specifically the detection of the hybridization signal, is developed so we do not have to set an intensity threshold for each colour channel. Intensity thresholds are often set based on the intensity of the spots in the negative control, which can introduce a bias. We have now added the intensity distributions of spots in the negative control and compared this to the intensity of “true” hybridization signal (see Supplementary Fig. 18c), pasted here for convenience.

The dashed line represents a hypothetical intensity threshold that would be set if our analysis pipeline would include a thresholding step. Instead, the algorithm does not consider the absolute value (i.e., as would be the case if we were thresholding) but the relative values of the local maxima within the image. It operates under the assumption that the hybridization signal is in the right end tail of the pixel distributions (this is demonstrated in SN2 Fig4), pasted here for convenience.

The red box is where the algorithm assumes the “true” signal to be present in an image where signal is expected. However, if no hybridization signal is present (i.e., in the negative controls) and the image is noisy (in particular for the green channel), even the negative control will have a small right tail. As a result, the algorithm will assume that these are hybridization signals. Some of these signals will (by chance) resemble a gaussian point spread function whereby the support vector machine indicates it is a signal. This however is not an issue when actual hybridization signals are present, because the right tail will shift to the right. This will occur because the intensity of true signal is significantly higher (see plot above, now new Supplementary Fig. 18c) and the “false” signal will no longer be detected. Notably, because we do not implement a threshold the image analysis software will always detect signal in the negative control images (as the reviewer pointed out), because there is a lack of higher intensity “true” signal present.

To illustrate this point, we introduced true hybridisation signal into the negative control (i.e., previously detected as “false” signal) in silico by merging a negative control image with an imaging round from a true ARTseq-FISH experiment. We subsequently compared the number of false positives detected in the negative control and merged image, expecting the number of

false positive signals to be substantially lower in the merged image. We performed this for the green channel as this has the highest detection of “false” signal (and the most overlap with true signal). The detection of “false” signal in the negative control decreases >6-fold when we introduce “true signal” (see Supplementary Fig. 18d), pasted here for convenience.

Therefore, the detected spot count in the negative control provided in the figure the reviewer refers to (now Supplementary Fig. 18b) would be >6-fold lower in the presence of true hybridisation signal. We have amended the Figure to reflect this (see Supplementary Fig. 18b):

Minor comments

Reviewer comment:

Lines 179-180 ‘Notably, while unphosphorylated RB shows low spot counts at a protein level, its mRNA and phospho-RB display higher spot count’ I do not not understand the significance / intent of this sentence here, since the authors after commented that one can not compare the absolute counts between species.

Response:

The sentence the reviewer highlighted was intended to emphasize that we cannot compare the absolute counts between species we have now reworded this sentence to read (see line 183):

“Notably, while RB shows low spot counts at a protein level, its mRNA and phospho-RB display higher spot counts (Fig. 3b-d), indicating that the difference in spot count between total RB and phospho-RB is caused by differences in antibody specificity. As

mentioned previously, similar to other omics methods that rely on antibodies^{33, 34}, ARTseq-FISH is dependent on the binding efficiency of primary antibodies (for proteins) and PLPs (for mRNAs). Therefore, we cannot compare absolute counts between species."

Extended Data Fig5 – formatting / spelling error ‘Rrepresentative max intensity and ...’

Response:

We thank the reviewer for catching this typo and have now amended the Extended Data Fig. 5 (current Supplementary Fig. 19).

Reviewer #3:

Reviewer comment:

In this manuscript Hu et al develop a new technique known as ARTseqFISH to simultaneously measure protein and mRNA abundances in situ at relatively high throughput (67 targets). They apply this technique to mouse embryonic stem cells (mESCs) cultured on micropatterns and use this to demonstrate that the differentiation is position dependent. They also suggest based on expression of cell cycle markers that the differentiation is caused by cell cycle changes. I am not an expert on spatial transcriptomics or high throughput protein measurements, so I cannot comment on whether the current technique represents an advance over existing methods and will largely restrict my comments to the biological system the authors address. Here, there are significant issues including misunderstandings of this phase of development and a number of conclusions which are not supported by the data. I detail these concerns below.

Response:

We thank the reviewer for thoroughly reading the manuscript. As outlined below, we have performed the suggested experiments as well as major revisions to the text and believe that these have significantly improved the manuscript.

Reviewer comment:

1. The statement of the authors that "cells continue to self-renew by progressing through the cell cycle, or terminally differentiate by exiting the cell cycle during the G1 phase" is incorrect in this context. No terminal differentiation occurs during the two day time window starting from pluripotency which the authors study and even cells which have exited pluripotency (e.g. primitive streak cells, mesoderm progenitors, neural progenitors etc) will continue to cycle. This is important because the dichotomy between cycling and differentiation is fundamental to how the authors frame their data but it does not apply in this context.

Response:

We apologize for the confusing statements in our manuscript and appreciate that we were not precise in our description. To make the conclusion more accurate based on the obtained data, we have now revised the text to focus on "exit from pluripotency" rather than "differentiation". Specifically, we:

- 1) Removed cell-fate decisions from the title.*
- 2) Altered the abstract to highlight that the study focuses on initial changes in mRNA and protein abundance during the first 48 hours of exit from pluripotency.*
- 3) Altered the abstract to indicate that changes in cell-cycle genes precede changes in cell-lineage genes and removed any conclusions about causation.*
- 4) Changed the first paragraph of the introduction to present the goals as: "Measuring mRNA and protein levels as mouse embryonic stem cells (mESCs) exit pluripotency will offer insights into how micropattern positioning affects early gene expression changes." (see line 34).*

- 5) *Changed the last paragraph of the introduction to read “The gene expression profiles of 67 targets were analysed during the initial 48 hours of differentiation, allowing us to capture relative changes in mRNA and protein abundance as mESCs exit pluripotency.” (see line 61).*
- 6) *Framed the interpretation of revised Fig. 4-7 in the context of exit from pluripotency.*

Reviewer comment:

2. The authors misrepresent what has been shown previously on micropatterns. Particularly the statements "it remains unknown how positioning on the micropattern impacts cellular decisions to ultimately define cell fate[9-15]. Specifically, the sequence of events that influence cell fate decisions on micropatterns and what factors drive the choice between self-renewal and differentiation remain unclear". While there is more to learn, the references the authors cite clearly establish that there is position-dependent differentiation on micropatterns (this is the main reason the system is interesting) and have begun to work out in detail the mechanisms of this differentiation in terms of the signals involved, how they change in time, and how they influence cell fates. Rather than insisting that little is known, the authors need to discuss the new knowledge generated by their study in light of the extensive work that has already been done on micropatterned systems.

Response:

We apologize for our choice of words, and acknowledge that an impressive amount of work has previously been done on micropatterned systems. We have therefore rephrased the main text to reflect this (see line 32), pasted here for convenience:

“Micropatterning has proven to be a powerful tool in unveiling position-dependent differentiation that recapitulates the spatial patterning of germ layers observed in early embryonic development¹⁰⁻¹⁷. Measuring mRNA and protein levels as mouse embryonic stem cells (mESCs) exit pluripotency will offer insights into how micropattern positioning affects early gene expression changes.”

Reviewer comment:

3. The discussion of Nodal in lines 213-216 is inaccurate. First, the antibody the authors use for "unphosphorylated" Smad2 (31H15L4) is actually an antibody for total Smad2, both phosphorylated and not. Thus, it should largely agree with mRNA distribution, and that fact that pSmad2 rather than Smad2 agrees with the mRNA distribution is likely the result of experimental error as the differences between edge and center in these factors are small. Moreover, it is not correct that unphosphorylated Smad has "bifunctional roles in the regulation of both self-renewal and differentiation" as Smad performs its primary functions when phosphorylated. It is the phosphorylated form that does this. This example shows the dangers of interpreting the data to fit an expected narrative.

Response:

We apologize for the unclear statement. In order to determine the experimental error of our data we have now quantified the fold-change localisation for 4 biological replicates (see new Fig. 4e). In addition, to interpret the data more holistically we altered Fig. 4e to display the fold-change localisation to the edge or centre of the micropattern for targets divided based on their function. We split these categories into lineage, pluripotency, cell cycle and other targets. Furthermore, to avoid selecting particular targets that fit an expected narrative, we altered the text to first describe these localisations based on the target class, rather than particular targets (see line 213):

“Overall, lineage, pluripotency as well as cell cycle associated proteins are more highly expressed in cells localised towards the edge of the micropattern. Specifically, lineage-associated proteins Brachyury and GATA6 are mostly enriched in cells at the edge of the micropattern (Fig. 4e and Supplementary Fig. 25a), as are the pluripotency proteins Nanog and Sox2. Surprisingly, some mRNAs (i.e., STAT3 and CDK2) appear localised to the centre although their (phospho-)protein counterparts are localised to the edge. This could either be because cells at the edge are translationally more active

and thereby require less mRNA to produce a higher amount of protein. Alternatively, cells could be in the process of up or down regulating their mRNA, and differences in mRNA and protein half-life could lead to a lack of correlation between the two species⁵⁶.”

Moreover, to be more accurate, we now explicitly state in the main text that the amount of protein is total unless otherwise specified (see line 174):

“When discussing the protein targets detected using the respective antibodies (Table S2), it is important to note that these measurements likely represent the total amount of protein detected, which includes both the unmodified and modified forms. Conversely, phosphorylated proteins were detected with specific antibodies (*protein or phos-protein in figures).”

We have also explained in more detail in the main text that the localisation analysis might miss more subtle position-dependent changes in expression of certain genes such as Smad2 and RB in cells at the centre compared to the edge (as previously shown in Supplementary Fig. 17b-c):

This difference is likely masked in the localisation analysis because, as the reviewer states, the changes in expression levels are subtle. We have now more clearly explained this in the text (see line 222):

“Lastly, when calculating the mean fold change in expression level in cells located in the centre versus edge, cells >125 μm from the edge were defined as being in the centre of the micropattern (Fig. 4c, bottom bar), which is possibly an oversimplification. For example, Smad2 and RB mRNA and protein are enriched in cells at the edge while our analysis suggests that the phosphorylated proteins are more homogeneously expressed across the micropattern (Fig. 4e). However, when quantifying the average expression profile of spots per cell at seven positions across the micropattern, a reduction in phospho-RB and phospho-Smad2 protein at the centre (~375 μm from the edge) of the micropattern becomes evident (Supplementary Fig. 25b-d). Consequently, the localisation analysis likely masks more subtle changes in gene expression between cells at the edge and centre of the micropattern.”

Reviewer comment:

4. There may be a similar issue with the authors discussion of phorsphorylation of Rb protein, but i cannot tell as no catalogue or clone number is given for the antibody. All antibodies should have manufacturers and catalogue numbers listed.

Response:

The Supplementary table (Table S2) already contained all the catalogue numbers for the antibodies. In addition, as mentioned above, in the main text (line 174) we now explicitly state that proteins are total unless marked as phosphorylated.

Reviewer comment:

5. The discussion of apoptosis and senescence is not supported by the data. p53 expression alone is not sufficient to conclude that apoptosis is occurring, the authors should analyze specific markers of this process. It is unclear what the claims regarding senescence are based on.

Response:

We thank the reviewer for this suggestion and have now taken several approaches to validate that apoptosis is occurring at the centre of the micropattern.

First, we performed additional analyses that specifically quantified nuclear shape based on an irregularity index (NII). When plotting the NII against nuclear area, it allows quantification of apoptosis among other cellular states⁴ (see Supplementary Fig. 27):

We observe that after 48 hours at the centre of the micropattern there appears an increase in apoptotic nuclei.

Moreover, we performed additional experiments to verify apoptosis with mid-, and late-stage apoptosis markers (Caspase-3 and 7-AAD respectively). The analysis reveals increased apoptosis, in particular late-stage apoptosis, in the centre of the micropattern at 48 hours after LIF withdrawal (see Supplementary Fig. 27).

To reflect these data, we have amended the main text (see line 394), to read:

“Caspase-3 and 7-AAD staining as well as nuclear morphology analysis indicate that this lack of proliferation could be connected to increased apoptosis at centre of the micropattern⁶³ (Supplementary Fig. 27).”

Lastly, although the reduced translational activity and reduced Ki67 detection (see new Fig. 6a-c and 6j respectively, as well as response to reviewer comment 6) is consistent with increased senescence in the centre of the micropattern, nuclear morphology analysis (see above) does not indicate increased senescence in the centre (or edge) of the micropattern. We have therefore removed the senescence argument from the text.

Reviewer comment:

6. The discussions of proliferation rate and cell density are also not warranted. First, one cannot make conclusions about proliferation rate from the expression data at alone. The authors need perform live cell imaging to actually determine the interval between cell divisions of individual cells. For example, the statement that "After 48 hours cells at the edge enter and remain in the G1 phase (Fig. 6h), from which they enter the terminal differentiation stage" is not warranted. All that is known from the data is a that a somewhat larger fraction of cells are in G1 at 48 hours. This does not necessarily result from terminal differentiation but may reflect a lengthening of the G1 phase of the cycle as cells exist pluripotency. In fact, as noted above, it is not possible for cells to go from pluripotency to terminal differentiation in 48 hours. Second, variations in cell density do not necessarily result from variations in proliferation but can also be due to cell movements. Both in vivo and some micropatterned systems it has been shown that cells migrate into a primitive streak like region and this will result in a local increase in density. Again, live cell imaging is required to distinguish these possibilities.

Response:

We thank the reviewer for these suggestions and have performed extensive validation experiments (new Fig. 6 and Fig. 7) to confirm that cells at the edge of the micropattern exhibit increased proliferation. We have also rewritten the text associated with Fig. 6 and 7, now entitled: “Cellular location on the micropattern impacts proliferation” (see line 358-402) and “Cellular location on the micropattern impacts cell cycling dynamics” (see line 422-466). The re-written section and figure are too large to paste here, however to summarize we have performed:

- 1) *Additional experiments where we exposed cells to the amino acid analogue O-propargyl-puromycin (OPP) and detected the incorporation of OPP into new synthesised proteins at different time points after LIF withdrawal. These experiments demonstrate reduced translational activity at the centre of the micropattern (Fig. 6a-c).*
- 2) *EdU staining (to detect actively dividing cells) revealing an increased number of dividing cells closer to the edge of the micropattern already after 12 hours of LIF withdrawal (Fig. 6h-i).*
- 3) *Ki67 staining (a marker for cell cycling) that shows higher intensity towards the edge of the micropattern than the centre (Fig. 6j).*

- 4) Time lapse microscopy experiments of Fucci mESCs^{5, 6} and single cell tracking demonstrating differences in the G1 phase when comparing cells at the edge and centre of the micropattern (Fig. 7a-g).
- 5) Single-cell tracking of both Nanog-GFP^{7, 8} and Fucci mESCs indicating minimal migration of cells (Supplementary Fig. 30).

Together these data point towards increased proliferation at the edge of the micropattern. Yet since high throughput single-cell tracking of micropatterned mESCs is very difficult, we cannot completely exclude fast migration of some cells. We have amended the text to reflect this (see line 462):

“While the potential for rapid cell migration that elude cell tracking exists, the prevailing data indicates that cells remain either at the centre or the edge of the micropattern (Supplementary Fig. 30c).”

Reviewer comment:

7. The conclusions about the temporal sequence of events over 48 hours are almost solely inferred from a pseudotime analysis. This is problematic in any case, but is especially problematic for cyclical events such as the cell cycle. In fact, contrary to the claims the authors make in the text, the expression of cell cycle genes in pseudotime is cyclic (fig 6d), not reflecting a steady progression from proliferation to exit from the cell cycle. The fact that the cell cycle genes appear cyclic here raises the possibility that the pseudotime in the authors calculations does not reflect a differentiation trajectory but is simply ordering cells by cell cycle phase. If the pseudotime actually reflects a differentiation trajectory, why do the self-renewal genes begin to increase again at the end?

Response:

To improve the time-resolution of the data, we have now performed a new ARTseq-FISH experiment which includes a 12-hour time-point. Furthermore, we plotted the relative changes in target abundance across the micropattern over time as a 2D heatmap (new Fig. 5c). To more accurately reflect the data and prevent any potential effects from the cell cycle on the pseudotime analysis, we replaced the pseudotime analysis with the expression levels of clustered targets 0, 12, 24, and 48 hours after LIF withdrawal in individual cells located at the centre or the edge (new Fig. 5d). Pasted here for convenience:

These data show that at the edge of the micropattern, changes in cell cycle genes precede changes in genes associated with endoderm and mesoderm lineages. We have amended the text regarding cell cycle genes to read (see line 285):

“To further explore whether proteins are expressed differently based on cellular location on the micropattern, we quantified gene expression profiles for individual cells situated either at the true edge or at the true centre of the micropattern (Fig. 5b). Once again, we categorized targets based on their functions and plotted the relative single-cell changes over the 0, 12, 24, and 48 hours of LIF withdrawal. At the micropattern's

edge, cells displayed an inverse behaviour between proteins related to lengthening the G1 phase (Fig. 5d, top: cell cycle 1) and cell cycle progression (Fig. 5d, top: cell cycle 2). Specifically, at 12 hours, we observed a decrease in proteins associated with cell cycle 1 (lengthening G1) and an increase in proteins related to cell cycle 2 (cell cycle progression). These findings suggested that within the first ~12 hours of LIF withdrawal, cells alter the expression of genes associated with cell cycle progression, potentially leading to higher proliferation rates during this time. In contrast, at the micropattern's centre, the levels of cell cycle 1 and 2 genes initially remain constant, and only after 12-24 hours of LIF withdrawal do protein levels of genes associated with lengthening the G1 phase increase (Fig. 5d, bottom). These results indicated that after 12 hours, the G1 phase of cells located at the centre might start increasing.”

To further discern the single-cell gene expression profiles at the edge of the micropattern at 48 hours of LIF withdrawal we generated a correlation matrix of protein targets associated with pluripotency, endoderm and mesoderm lineages (see Fig. 5g), pasted here for convenience:

We have amended the text to read (see line 331):

“Interestingly, at the single-cell level, we found a negative correlation between pluripotency-associated genes (i.e., OCT4 and Sox2) and lineage-specific markers (Fig. 5g) in cells at the edge of the micropattern. Furthermore, we observed the emergence of distinct clusters, with CDX2 (Trophectoderm) segregating separately from mesodermal markers (i.e., Brachyury, and N-Cadherin), as well as endodermal markers (i.e., GATA6 and Sox17).

Lastly, the increase of self-renewal targets at the very edge of the micropattern is consistent with EdU staining performed, which indicates that while EdU incorporation (a measure of the number of cells in the S phase) decreases across the whole micropattern, there is a very thin boarder of cells at the very edge, where EdU incorporation is significantly higher (see Fig. 6h-i), pasted here for convenience.

Reviewer comment:

8. Even if one accepts the authors claims regarding pseudotime, the conclusion that the differentiation events result from cell cycle changes is not warranted. first, some of the differentiation genes such as

Snail and Sox17 change as rapidly as the cell cycle genes. Second, even if cell cycle genes do change before differentiation ones, perturbative experiments would be required to establish causation.

Response:

We agree with the reviewer that arguments of causation would require perturbation experiments. We have therefore removed all conclusions regarding causation and instead explain that changes in lineage associated genes are preceded by changes in cell cycle genes. These changes are incorporated in the abstract, the end of the introduction (line 65), as well as the re-written section associated with the new Fig. 5 (starting line 266).

Reviewer comment:

9. It isn't clear to me whether the changes in differentiation markers are biologically significant. For example, Bra seems to change about 2 fold over pseudotime but one would expect a much larger increase in actual primitive streak. The authors should include positive controls (such as established mesoderm differentiation protocols) and quantify the expression of differentiation genes relative to expression in these controls.

Since we now primarily focus on exit from pluripotency we performed two sets of additional validation experiments to address the reviewer's concerns. First, we compared the fold-change over time in Nanog protein at the edge and centre of the micropattern as quantified by ARTseq-FISH to live-cell imaging of a micropatterned Nanog-GFP mESC line imaged for 46.5 hours after LIF withdrawal. These data reflect similar trends in Nanog expression over time (see Fig. 5e), pasted here for convenience:

We have amended the text to read (see line 315):

“While the overall trend is consistent with ARTseq-FISH Nanog data (Fig 5e, top), the initial decrease observed in the time-lapse experiment is not visible by ARTseq-FISH. This could be because ARTseq-FISH reveals expression levels of cells that are fixed after 0, 12, 24, and 48 hours post LIF withdrawal and do not represent a true time-course, which might conceal rapid changes in expression levels. Alternatively, it is possible that the differences between the time-lapse experiment and ARTseq-FISH are cell-type specific.”

Furthermore, we quantified OCT4 abundance by conventional immunofluorescence and find that the changes observed with ARTseq-FISH are equally subtle when measured by immunofluorescence (see Fig. 5f), pasted here for convenience:

Together these data indicate that the subtle changes in gene expression measured by ARTseq-FISH are reproducible by other methods.

The text now reads (see line 321):

“Therefore, we sought to confirm the expression of another pluripotency marker (OCT4) by immunofluorescence at 0 and 48 hours of LIF withdrawal, which shows a similar subtle trend as ARTseq-FISH (Fig. 5f). Together, these data indicate that cells at the centre and edge of the micropattern show subtle changes in expression of pluripotency genes that occur over time.”

Reviewer comment:

10. It seems that cell segmentation is performed using a nuclear image, but many of the targets will be cytoplasmic. The authors do not have a definitive way of associating each spot with the correct nucleus without marking the cell boundaries. This source of error should be acknowledged.

We thank the reviewer for bringing this up and have now included this in the text (see line 148), pasted here for convenience:

“Notably, we assign each target to a specific cell based on the shortest distance to the nucleus, hence there is a possibility of incorrect assignments if the distance from the nucleus to the cell membrane varies significantly between neighbouring cells.”

References

- Shah, S.; Takei, Y.; Zhou, W.; Lubeck, E.; Yun, J.; Eng, C.-H. L.; Koulina, N.; Cronin, C.; Karp, C.; Liaw, E. J.; Amin, M.; Cai, L., Dynamics and Spatial Genomics of the Nascent Transcriptome by Intron seqFISH. *Cell* **2018**, *174* (2), 363-376.e16.
- Xia, C.; Fan, J.; Emanuel, G.; Hao, J.; Zhuang, X., Spatial transcriptome profiling by MERFISH reveals subcellular RNA compartmentalization and cell cycle-dependent gene expression. *Proceedings of the National Academy of Sciences* **2019**, *116* (39), 19490-19499.
- Kramer, B. A.; Sarabia del Castillo, J.; Pelkmans, L., Multimodal perception links cellular state to decision-making in single cells. *Science* **2022**, *377* (6606), 642-648.
- Filippi-Chiela, E. C.; Oliveira, M. M.; Jurkovski, B.; Callegari-Jacques, S. M.; Silva, V. D. d.; Lenz, G., Nuclear Morphometric Analysis (NMA): Screening of Senescence, Apoptosis and Nuclear Irregularities. *PLOS ONE* **2012**, *7* (8), e42522.
- Ter Huurne, M.; Chappell, J.; Dalton, S.; Stunnenberg, H. G., Distinct Cell-Cycle Control in Two Different States of Mouse Pluripotency. *Cell Stem Cell* **2017**, *21* (4), 449-455.e4.
- ter Huurne, M.; Peng, T. R.; Yi, G. Q.; van Mierlo, G.; Marks, H.; Stunnenberg, H. G., Critical Role for P53 in Regulating the Cell Cycle of Ground State Embryonic Stem Cells. *Stem Cell Rep* **2020**, *14* (2), 175-183.

7. Desai, R. V.; Chen, X.; Martin, B.; Chaturvedi, S.; Hwang, D. W.; Li, W.; Yu, C.; Ding, S.; Thomson, M.; Singer, R. H.; Coleman, R. A.; Hansen, M. M. K.; Weinberger, L. S., A DNA repair pathway can regulate transcriptional noise to promote cell fate transitions. *Science* **2021**, *373* (6557), eabc6506.
8. Sokolik, C.; Liu, Y.; Bauer, D.; McPherson, J.; Broeker, M.; Heimberg, G.; Qi, Lei S.; Sivak, David A.; Thomson, M., Transcription Factor Competition Allows Embryonic Stem Cells to Distinguish Authentic Signals from Noise. *Cell Syst* **2015**, *1* (2), 117-129.

Reviewer #1 (Remarks to the Author):

The manuscript has improved by down-tuning a lot of the previously unsubstantiated claims, but I am still not convinced about the the claims on specificity and sensitivity of the method based on the provided data.

Reviewer #4 (Remarks to the Author):

Hu et al present a technique to detect the expression of multiple mRNA and proteins in micropatterned colonies of mESCs. This technique addresses a challenge in the fields of in vitro differentiation assays or embryology which is the characterization of the heterogeneity and spatial organization emerging during these processes.

Cell identity can be challenging to assess as it often requires multiple markers to make a clear conclusion. This can be challenging as good antibodies are not always available and at most 3-4 can be detected on the same sample. Therefore being able to detect and assess the co-expression of tens of RNA molecules and proteins on the same sample would be indeed extremely useful.

Like reviewer#3, I am not an expert in spatial transcriptomics methods, so I won't comment that part/ reviewer#1 gave feedback on this aspect.

My impression is that the author response's to individual comments of reviewer#3 is satisfactory. They have brought precisions and when necessary, tone down unsupported claims or provided additional evidence supporting them. See comment on individual comments in red in the text bellow.

However the more general comment of reviewer#3 that there is a lot on confusion about the biology of the studied system is still not fully corrected.

For instance:

Line 334 : "we observed the emergence of distinct clusters, with CDX2 (Trophectoderm) segregating separately from mesodermal markers (i.e., Brachyury, and N-Cadherin),"

At the stage modelled by this experiment (peri gastrulation), CDX2 is expressed in many cell types. Mostly derivating from the posterior primitive streak like the different types of extraembryonic mesoderm but not trophectoderm which is a cell type present in earlier stages embryos. It is interesting to see that in table S5 CDX2 is assigned to trophectoderm citing the Morgani et al paper that explains really clearly the identity of the different cell types expressing CDX2 can be posterior epiblast, extraembryonic mesoderm ACD or AOM, .

Same goes with GATA6 which is assigned to endoderm. Again the morgani et al paper makes it clear that GATA6 can be present in many cell types at that stage. Only careful analysis of co-expression with other markers (OTX2, SOX17, BRA, CDX2) to properly assign an identity to those cells (mesoderm 1, endoderm, PS or allantois respectively)

Endoderm cells co-express FOXA2 and SOX17. It is not clear what markers or combination of markers is used to assign the endoderm identity to a cell in the endoderm graph of figure 5c,d to for instance

ARTseqFISH is by design a method capable of measuring co-expression of multiple markers in a single cell, it is a bit disappointing that this capability is not fully exploited in the present work do precisely assign cell identity. It could have been used to detect the presence of PGCs of notochord cells for instance (although BMP or WNT stimulation might be necessary to have those cell type appear)

Additional comment:

All over the manuscript, the number of experiments or colonies used in each figure is not always clear. For instance in fig 4abc. Is data presented for one colony only? This might be OK for panels a

and b but for panel c it might be stronger to show localization data averaged over many colonies and full biological replicates. Same for fig5cd; is this data from 1 colony or an average of many colonies? The number of samples should be stated in figure legends .

Additional remarks on Reviewer #3 comments:

Reviewer comment:

In this manuscript Hu et al develop a new technique known as ARTseqFISH to simultaneously measure protein and mRNA abundance in situ at relatively high throughput (67 targets). They apply this technique to mouse embryonic stem cells (mESCs) cultured on micropatterns and use this to demonstrate that the differentiation is position dependent. They also suggest based on expression of cell cycle markers that the differentiation is caused by cell cycle changes. I am not an expert on spatial transcriptomics or high throughput protein measurements, so I cannot comment on whether the current technique represents an advance over existing methods and will largely restrict my comments to the biological system the authors address. Here, there are significant issues including misunderstandings of this phase of development and a number of conclusions which are not supported by the data. I detail these concerns below.

Response:

We thank the reviewer for thoroughly reading the manuscript. As outlined below, we have performed the suggested experiments as well as major revisions to the text and believe that these have significantly improved the manuscript.

Reviewer comment:

1. The statement of the authors that "cells continue to self-renew by progressing through the cell cycle, or terminally differentiate by exiting the cell cycle during the G1 phase" is incorrect in this context. No terminal differentiation occurs during the two day time window starting from pluripotency which the authors study and even cells which have exited pluripotency (e.g. primitive streak cells, mesoderm progenitors, neural progenitors etc) will continue to cycle. This is important because the dichotomy between cycling and differentiation is fundamental to how the authors frame their data but it does not apply in this context.

Response:

We apologize for the confusing statements in our manuscript and appreciate that we were not precise in our description. To make the conclusion more accurate based on the obtained data, we have now revised the text to focus on "exit from pluripotency" rather than "differentiation".

Specifically, we:

- 1) Removed cell-fate decisions from the title.
- 2) Altered the abstract to highlight that the study focuses on initial changes in mRNA and protein abundance during the first 48 hours of exit from pluripotency.
- 3) Altered the abstract to indicate that changes in cell-cycle genes precede changes in cell-lineage genes and removed any conclusions about causation.
- 4) Changed the first paragraph of the introduction to present the goals as: "Measuring mRNA and protein levels as mouse embryonic stem cells (mESCs) exit pluripotency will offer insights into how micropattern positioning affects early gene expression changes." (see line 34).
- 5) Changed the last paragraph of the introduction to read "The gene expression profiles of 67 targets were analysed during the initial 48 hours of differentiation, allowing us to capture relative changes in mRNA and protein abundance as mESCs exit pluripotency." (see line 61).
- 6) Framed the interpretation of revised Fig. 4-7 in the context of exit from pluripotency.

additional reviewer comment : I think the proposed corrections answer the concerns of the referee

Reviewer comment:

2. The authors misrepresent what has been shown previously on micropatterns. Particularly the statements "it remains unknown how positioning on the micropattern impacts cellular decisions to ultimately define cell fate[9-15]. Specifically, the sequence of events that influence cell fate

decisions on micropatterns and what factors drive the choice between self-renewal and differentiation remain unclear". While there is more to learn, the references the authors cite clearly establish that there is position-dependent differentiation on micropatterns (this is the main reason the system is interesting) and have begun to work out in detail the mechanisms of this differentiation in terms of the signals involved, how they change in time, and how they influence cell fates. Rather than insisting that little is known, the authors need to discuss the new knowledge generated by their study in light of the extensive work that has already been done on micropatterned systems.

Response:

We apologize for our choice of words, and acknowledge that an impressive amount of work has previously been done on micropatterned systems. We have therefore rephrased the main text to reflect this (see line 32), pasted here for convenience:

"Micropatterning has proven to be a powerful tool in unveiling position-dependent differentiation that recapitulates the spatial patterning of germ layers observed in early embryonic development¹⁰⁻¹⁷. Measuring mRNA and protein levels as mouse embryonic stem cells (mESCs) exit pluripotency will offer insights into how micropattern positioning affects early gene expression changes."

additional reviewer comment : OK the proposed change addresses the reviewer's question in a satisfactory manner

Reviewer comment:

3. The discussion of Nodal in lines 213-216 is inaccurate. First, the antibody the authors use for unphosphorylated Smad2 (31H15L4) is actually an antibody for total Smad2, both phosphorylated and not. Thus, it should largely agree with mRNA distribution, and that fact that pSmad2 rather than Smad2 agrees with the mRNA distribution is likely the result of experimental error as the differences between edge and center in these factors are small. Moreover, it is not correct that unphosphorylated Smad has "bifunctional roles in the regulation of both self-renewal and differentiation" as Smad performs its primary functions when phosphorylated. It is the phosphorylated form that does this. This example shows the dangers of interpreting the data to fit an expected narrative.

Response:

We apologize for the unclear statement. In order to determine the experimental error of our data we have now quantified the fold-change localisation for 4 biological replicates (see new Fig. 4e). In addition, to interpret the data more holistically we altered Fig. 4e to display the fold-change localisation to the edge or centre of the micropattern for targets divided based on their function. We split these categories into lineage, pluripotency, cell cycle and other targets. Furthermore, to avoid selecting particular targets that fit an expected narrative, we altered the text to first describe these localizations based on the target class, rather than particular targets (see line 213):

additional reviewer comment :

OK it's good to analyse 4 biological replicates but it would be better to be a bit more precise. What is meant by biological replicates? How many colonies were considered by replicates. This info should be in the legend of the figure

"Overall, lineage, pluripotency as well as cell cycle associated proteins are more highly expressed in cells localised towards the edge of the micropattern. Specifically, lineage-associated proteins Brachyury and GATA6 are mostly enriched in cells at the edge of the micropattern (Fig. 4e and Supplementary Fig. 25a), as are the pluripotency proteins Nanog and Sox2. Surprisingly, some mRNAs (i.e., STAT3 and CDK2) appear localised to the centre although their (phospho-)protein counterparts are localised to the edge. This could either be because cells at the edge are translationally more active and

thereby require less mRNA to produce a higher amount of protein. Alternatively, cells could be in the process of up or down regulating their mRNA, and differences in mRNA and protein half-life could lead to a lack of correlation between the two species⁵⁶."

The statement "lineage-associated proteins Brachyury and GATA6 are mostly enriched in cells at the edge of the micropattern (Fig. 4e and Supplementary Fig. 25a), as are the pluripotency proteins Nanog and Sox2" is puzzling to me. Cells are both pluripotent and differentiated at the edge of the colony?

Is this over expression in the same cells? What are those cells then?

The description of marker co-expression should be done at the single cell level (see my comment in intro)

Moreover, to be more accurate, we now explicitly state in the main text that the amount of protein is total unless otherwise specified (see line 174):

"When discussing the protein targets detected using the respective antibodies (Table S2), it is important to note that these measurements likely represent the total amount of protein detected, which includes both the unmodified and modified forms. Conversely, phosphorylated proteins were detected with specific antibodies (*protein or phos-protein in figures)."

additional reviewer comment : OK

We have also explained in more detail in the main text that the localisation analysis might miss more subtle position-dependent changes in expression of certain genes such as Smad2 and RB in cells at the centre compared to the edge (as previously shown in Supplementary Fig. 17b-c): This difference is likely masked in the localisation analysis because, as the reviewer states, the changes in expression levels are subtle. We have now more clearly explained this in the text (see line 222): "Lastly, when calculating the mean fold change in expression level in cells located in the centre versus edge, cells >125 µm from the edge were defined as being in the centre of the micropattern (Fig. 4c, bottom bar), which is possibly an oversimplification. For example, Smad2 and RB mRNA and protein are enriched in cells at the edge while our analysis suggests that the phosphorylated proteins are more homogeneously expressed across the micropattern (Fig. 4e). However, when quantifying the average expression profile of spots per cell at seven positions across the micropattern, a reduction in phospho-RB and phospho-Smad2 protein at the centre (~375µm from the edge) of the micropattern becomes evident (Supplementary Fig. 25b-d). Consequently, the localisation analysis likely masks more subtle changes in gene expression between cells at the edge and centre of the micropattern."

Reviewer comment:

4. There may be a similar issue with the authors discussion of phorsphorylation of Rb protein, but i cannot tell as no catalogue or clone number is given for the antibody. All antibodies should have manufacturers and catalogue numbers listed.

Page 7 of 13

Response:

The Supplementary table (Table S2) already contained all the catalogue numbers for the antibodies. In addition, as mentioned above, in the main text (line 174) we now explicitly state that proteins are total unless marked as phosphorylated.

additional reviewer comment : OK

Reviewer comment:

5. The discussion of apoptosis and senescence is not supported by the data. p53 expression alone is not sufficient to conclude that apoptosis is occurring, the authors should analyze specific markers of this process. It is unclear what the claims regarding senescence are based on.

Response:

We thank the reviewer for this suggestion and have now taken several approaches to validate that apoptosis is occurring at the centre of the micropattern.

First, we performed additional analyses that specifically quantified nuclear shape based on an irregularity index (NII). When plotting the NII against nuclear area, it allows quantification of

apoptosis among other cellular states⁴ (see Supplementary Fig. 27):

We observe that after 48 hours at the centre of the micropattern there appears an increase in apoptotic nuclei. Moreover, we performed additional experiments to verify apoptosis with mid-, and late-stage

apoptosis markers (Caspase-3 and 7-AAD respectively). The analysis reveals increased apoptosis, in particular late-stage apoptosis, in the centre of the micropattern at 48 hours after LIF withdrawal (see Supplementary Fig. 27).

OK these additional markers make the claim more supported

To reflect these data, we have amended the main text (see line 394), to read:

"Caspase-3 and 7-AAD staining as well as nuclear morphology analysis indicate that this lack of proliferation could be connected to increased apoptosis at centre of the micropattern⁶³ (Supplementary Fig. 27)."

Lastly, although the reduced translational activity and reduced Ki67 detection (see new Fig. 6a-c and 6j respectively, as well as response to reviewer comment 6) is consistent with increased senescence in the centre of the micropattern, nuclear morphology analysis (see above) does not indicate increased senescence in the centre (or edge) of the micropattern. We have therefore removed the senescence argument from the text.

additional reviewer comment : OK

Reviewer comment:

6. The discussions of proliferation rate and cell density are also not warranted. First, one cannot make

conclusions about proliferation rate from the expression data at alone. The authors need perform live

cell imaging to actually determine the interval between cell divisions of individual cells. For example,

the statement that "After 48 hours cells at the edge enter and remain in the G1 phase (Fig. 6h), from

which they enter the terminal differentiation stage" is not warranted. All that is known from the data is a that a somewhat larger fraction of cells are in G1 at 48 hours. This does not necessarily result from

terminal differentiation but may reflect a lengthening of the G1 phase of the cycle as cells exist pluripotency. In fact, as noted above, it is not possible for cells to go from pluripotency to terminal differentiation in 48 hours. Second, variations in cell density do not necessarily result from variations

in proliferation but can also be due to cell movements. Both in vivo and some micropatterned systems

it has been shown that cells migrate into a primitive streak like region and this will result in a local increase in density. Again, live cell imaging is required to distinguish these possibilities.

Response:

We thank the reviewer for these suggestions and have performed extensive validation experiments (new Fig. 6 and Fig. 7) to confirm that cells at the edge of the micropattern exhibit increased proliferation. We have also rewritten the text associated with Fig. 6 and 7, now entitled: "Cellular location on the micropattern impacts proliferation" (see line 358-402) and "Cellular location on the micropattern impacts cell cycling dynamics" (see line 422-466). The re-written section and figure are too large to paste here, however to summarize we have performed:

1) Additional experiments where we exposed cells to the amino acid analogue Opropargyl-puromycin (OPP) and detected the incorporation of OPP into new synthesised proteins at different time points after LIF withdrawal. These experiments demonstrate reduced translational activity at the centre of the

micropattern (Fig. 6a-c). 2) EdU staining (to detect actively dividing cells) revealing an increased number of dividing cells closer to the edge of the micropattern already after 12 hours of LIF withdrawal (Fig. 6h-i).

3) Ki67 staining (a marker for cell cycling) that shows higher intensity towards the edge of the micropattern than the centre (Fig. 6j). Page 9 of 13

4) Time lapse microscopy experiments of Fucci mESCs^{5, 6} and single cell tracking demonstrating differences in the G1 phase when comparing cells at the edge and centre of the micropattern (Fig.

7a-g).

5) Single-cell tracking of both Nanog-GFP7, 8 and Fucci mESCs indicating minimal migration of cells (Supplementary Fig. 30).

Together these data point towards increased proliferation at the edge of the micropattern. Yet since high throughput single-cell tracking of micropatterned mESCs is very difficult, we cannot completely exclude fast migration of some cells. We have amended the text to reflect this (see line 462):

"While the potential for rapid cell migration that elude cell tracking exists, the prevailing data indicates that cells remain either at the centre or the edge of the micropattern (Supplementary Fig. 30c)."

additional reviewer comment : OK this additional data now supports the claim that proliferation is increased at the edge compared to the center of colonies. The overall decrease in proliferation over time (fig 6h) is also noted in the text

Reviewer comment:

7. The conclusions about the temporal sequence of events over 48 hours are almost solely inferred from a pseudotime analysis. This is problematic in any case, but is especially problematic for cyclical events such as the cell cycle. In fact, contrary to the claims the authors make in the text , the expression of cell cycle genes in pseudotime is cyclic (fig 6d), not reflecting a steady progression from proliferation to exit from the cell cycle. The fact that the cell cycle genes appear cyclic here raises the possibility that

the pseudotime in the authors calculations does not reflect a differentiation trajectory but is simply ordering cells by cell cycle phase. If the pseudotime actually reflects a differentiation trajectory, why

do the self-renewal genes begin to increase again at the end?

Response:

To improve the time-resolution of the data, we have now performed a new ARTseq-FISH experiment which includes a 12-hour time-point. Furthermore, we plotted the relative changes in target abundance across the micropattern over time as a 2D heatmap (new Fig. 5c). To more accurately reflect the data and prevent any potential effects from the cell cycle on the pseudotime analysis, we replaced the pseudotime analysis with the expression levels of clustered targets 0, 12, 24, and 48 hours after LIF withdrawal in individual cells located at the centre or the edge (new Fig. 5d). Pasted here for convenience:

These data show that at the edge of the micropattern, changes in cell cycle genes precede changes in genes associated with endoderm and mesoderm lineages. We have amended the text regarding cell cycle genes to read (see line 285): "To further explore whether proteins are expressed differently based on cellular location on the micropattern, we quantified gene expression profiles for individual cells situated either at the true edge or at the true centre of the micropattern (Fig. 5b). Once

again, we categorized targets based on their functions and plotted the relative singlecell changes over the 0, 12, 24, and 48 hours of LIF withdrawal. At the micropattern's edge, cells displayed an inverse behaviour between proteins related to lengthening the G1 phase (Fig. 5d, top: cell cycle 1) and cell cycle progression (Fig. 5d, top: cell cycle 2). Specifically, at 12 hours, we observed a decrease in proteins associated with cell cycle 1 (lengthening G1) and an increase in proteins related to cell cycle 2 (cell cycle progression). These findings suggested that within the first ~12 hours of LIF withdrawal, cells alter the expression of genes associated with cell cycle progression, potentially leading to higher proliferation rates during this time. In contrast, at the micropattern's centre, the levels of cell cycle 1 and 2 genes initially remain constant, and only after 12-24 hours of LIF withdrawal do protein levels of genes associated with lengthening the G1 phase increase (Fig. 5d, bottom). These results indicated that after 12 hours, the G1 phase of cells located at the centre might start increasing."

To further discern the single-cell gene expression profiles at the edge of the micropattern at 48 hours of LIF withdrawal we generated a correlation matrix of protein targets associated with pluripotency, endoderm and mesoderm lineages (see Fig. 5g), pasted here for convenience: We have amended the text to read (see line 331):

“Interestingly, at the single-cell level, we found a negative correlation between pluripotency-associated genes (i.e., OCT4 and Sox2) and lineage-specific markers (Fig. 5g) in cells at the edge of the micropattern. Furthermore, we observed the emergence of distinct clusters, with CDX2 (Trophoectoderm) segregating separately from mesodermal markers (i.e., Brachyury, and N-Cadherin), as well as endodermal markers (i.e., GATA6 and Sox17).

Lastly, the increase of self-renewal targets at the very edge of the micropattern is consistent with EdU staining performed, which indicates that while EdU incorporation (a measure of the number of cells in the S phase) decreases across the whole micropattern, there is a very thin boarder of cells at the very edge, where EdU incorporation is significantly higher (see Fig. 6hi), pasted here for convenience.

Reviewer comment:

8. Even if one accepts the authors claims regarding pseudotime, the conclusion that the differentiation events result from cell cycle changes is not warranted. first, some of the differentiation genes such as Page 11 of 13 Snail and Sox17 change as rapidly as the cell cycle genes. Second, even if cell cycle genes do change before differentiation ones, perturbative experiments would be required to establish causation.

Response:

We agree with the reviewer that arguments of causation would require perturbation experiments. We have therefore removed all conclusions regarding causation and instead explain that changes in lineage associated genes are preceded by changes in cell cycle genes. These changes are incorporated in the abstract, the end of the introduction (line 65), as well as the re-written section associated with the new Fig. 5 (starting line 266).

additional reviewer comment : OK

Reviewer comment:

9. It isn't clear to me whether the changes in differentiation markers are biologically significant. For example, Bra seems to change about 2 fold over pseudotime but one would expect a much larger increase in actual primitive streak. The authors should include positive controls (such as established mesoderm differentiation protocols) and quantify the expression of differentiation genes relative to expression in these controls.

Since we now primarily focus on exit from pluripotency we performed two sets of additional validation experiments to address the reviewer's concerns. First, we compared the fold-change over time in Nanog protein at the edge and centre of the micropattern as quantified by ARTseq-FISH to live-cell imaging of a micropatterned Nanog-GFP mESC line imaged for 46.5 hours after LIF withdrawal. These data reflect similar trends in Nanog expression over time (see Fig. 5e), pasted here for convenience:

We have amended the text to read (see line 315):

“While the overall trend is consistent with ARTseq-FISH Nanog data (Fig 5e, top), the initial decrease observed in the time-lapse experiment is not visible by ARTseq-FISH. This could be because ARTseq-FISH reveals expression levels of cells that are fixed after 0, 12, 24, and 48 hours post LIF withdrawal and do not represent a true timecourse, which might conceal rapid changes in expression levels. Alternatively, it is possible that the differences between the time-lapse experiment and ARTseq-FISH are cell-type specific.”

Furthermore, we quantified OCT4 abundance by conventional immunofluorescence and find that the changes observed with ARTseq-FISH are equally subtle when measured by immunofluorescence (see Fig. 5f), pasted here for convenience:

Page 12 of 13

Together these data indicate that the subtle changes in gene expression measured by ARTseq-FISH are reproducible by other methods.

The text now reads (see line 321):

"Therefore, we sought to confirm the expression of another pluripotency marker (OCT4) by immunofluorescence at 0 and 48 hours of LIF withdrawal, which shows a similar subtle trend as ARTseq-FISH (Fig. 5f). Together, these data indicate that cells at the centre and edge of the micropattern show subtle changes in expression of pluripotency genes that occur over time."

additional reviewer comment :

OK but proper negative controls could have been added. They are not always possible but for some markers they can be easy.

For instance for phospho-SMAD2 using the SB43 inhibitor as a negative control and activin stimulation as a positive one give a lower and upper bound of the signal.

Isn't it puzzling that IF and ArtseqFISH data don't agree in fig 5f? significance of the differences at 0 and 48hr should be assessed. I don't agree that the difference is subtle for artseqfish at 0 vs 48h while it doesn't seem significant in IF

Reviewer comment:

10. It seems that cell segmentation is performed using a nuclear image, but many of the targets will be

cytoplasmic. The authors do not have a definitive way of associating each spot with the correct nucleus

without marking the cell boundaries. This source of error should be acknowledged.

We thank the reviewer for bringing this up and have now included this in the text (see line 148), pasted here for convenience:

"Notably, we assign each target to a specific cell based on the shortest distance to the nucleus, hence there is a possibility of incorrect assignments if the distance from the nucleus to the cell membrane varies significantly between neighbouring cells."

additional reviewer comment :

I agree with this concern and I think the solution chosen is not fully satisfying given the density of the tissue. A segmentation based on plasma membrane would be better in that case but I concede that this is more difficult to achieve. Maybe keeping only the signal overlapping with nuclei would minimize the possibility of assignment mistake?

References

1. Shah, S.; Takei, Y.; Zhou, W.; Lubeck, E.; Yun, J.; Eng, C.-H. L.; Koulina, N.; Cronin, C.; Karp, C.; Liaw, E. J.; Amin, M.; Cai, L., Dynamics and Spatial Genomics of the Nascent Transcriptome by Intron seqFISH. *Cell* 2018, 174 (2), 363-376.e16.
 2. Xia, C.; Fan, J.; Emanuel, G.; Hao, J.; Zhuang, X., Spatial transcriptome profiling by MERFISH reveals subcellular RNA compartmentalization and cell cycle-dependent gene expression. *Proceedings of the National Academy of Sciences* 2019, 116 (39), 19490-19499.
 3. Kramer, B. A.; Sarabia del Castillo, J.; Pelkmans, L., Multimodal perception links cellular state to decision-making in single cells. *Science* 2022, 377 (6606), 642-648.
 4. Filippi-Chiela, E. C.; Oliveira, M. M.; Jurkovski, B.; Callegari-Jacques, S. M.; Silva, V. D. d.; Lenz, G., Nuclear Morphometric Analysis (NMA): Screening of Senescence, Apoptosis and Nuclear Irregularities. *PLOS ONE* 2012, 7 (8), e42522.
 5. Ter Huurne, M.; Chappell, J.; Dalton, S.; Stunnenberg, H. G., Distinct Cell-Cycle Control in Two Different States of Mouse Pluripotency. *Cell Stem Cell* 2017, 21 (4), 449-455.e4.
 6. ter Huurne, M.; Peng, T. R.; Yi, G. Q.; van Mierlo, G.; Marks, H.; Stunnenberg, H. G., Critical Role for P53 in Regulating the Cell Cycle of Ground State Embryonic Stem Cells. *Stem Cell Rep* 2020, 14 (2), 175-183.
- Page 13 of 13
7. Desai, R. V.; Chen, X.; Martin, B.; Chaturvedi, S.; Hwang, D. W.; Li, W.; Yu, C.; Ding, S.; Thomson, M.; Singer, R. H.; Coleman, R. A.; Hansen, M. M. K.; Weinberger, L. S., A DNA repair pathway can regulate transcriptional noise to promote cell fate transitions. *Science* 2021, 373 (6557), eabc6506.
 8. Sokolik, C.; Liu, Y.; Bauer, D.; McPherson, J.; Broecker, M.; Heimberg

Reviewer 4 comments:

Overall comment:

Hu et al present a technique to detect the expression of multiple mRNA and proteins in micropatterned colonies of mESCs. This technique addresses a challenge in the fields of in vitro differentiation assays or embryology which is the characterization of the heterogeneity and spatial organization emerging during these processes. Cell identity can be challenging to assess as it often requires multiple markers to make a clear conclusion. This can be challenging as good antibodies are not always available and at most 3-4 can be detected on the same sample. Therefore being able to detect and assess the co-expression of tens of RNA molecules and proteins on the same sample would be indeed extremely useful.

Like reviewer#3, I am not an expert in spatial transcriptomics methods, so I won't comment that part/ reviewer#1 gave feedback on this aspect. My impression is that the author response's to individual comments of reviewer#3 is satisfactory. They have brought precisions and when necessary, tone down unsupported claims or provided additional evidence supporting them. See comment on individual comments in red in the text below. However the more general comment of reviewer#3 that there is a lot on confusion about the biology of the studied system is still not fully corrected.

Response:

We thank the reviewer for the thorough assessment of the manuscript. We have now addressed all of Reviewer 4's comments. In particular, we have addressed the comments which were raised from our response to Reviewer 3's original comments (see section entitled: "Reviewer 4 comments related to Reviewer 3's comments") where we have kept the numbering the same as the original response, yet have removed those comments that Reviewer 4 deemed adequately addressed. The additional points raised by Reviewer 4 we have also addressed as outlined below.

Reviewer 4, comment 1:

Line 334 : "we observed the emergence of distinct clusters, with CDX2 (Trophectoderm) segregating separately from mesodermal markers (i.e., Brachyury, and N-Cadherin),"

At the stage modelled by this experiment (peri gastrulation), CDX2 is expressed in many cell types. Mostly derivating from the posterior primitive streak like the different types of extraembryonic mesoderm but not trophectoderm which is a cell type present in earlier stages embryos. It is interesting to see that in table S5 CDX2 is assigned to trophectoderm citing the Morgani et al paper that explains really clearly the identity of the different cell types expressing CDX2 can be posterior epiblast, extraembryonic mesoderm ACD or AOM.

Same goes with GATA6 which is assigned to endoderm. Again the morgani et al paper makes it clear that GATA6 can be present in many cell types at that stage. Only careful analysis of co-expression with other markers (OTX2, SOX17, BRA, CDX2) to properly assign an identity to those cells (mesoderm 1, endoderm, PS or allantois respectively)

Endoderm cells co-express FOXA2 and SOX17. It is not clear what markers or combination of markers is used to assign the endoderm identity to a cell in the endoderm graph of figure 5c,d to for instance

Response:

In order to avoid any confusion, we have now removed the description of cell identity and only described the expression of particular proteins or mRNAs.

Reviewer 4, comment 2:

ARTseqFISH is by design a method capable of measuring co-expression of multiple markers in a single cell, it is a bit disappointing that this capability is not fully exploited in the present work do precisely assign

cell identity. It could have been used to detect the presence of PGCs or notochord cells for instance (although BMP or WNT stimulation might be necessary to have those cell type appear)

Response

We thank the reviewer for bringing this up and have now included UMAP analysis data in Figure 4f. Although slight, the UMAP analysis shows separation of cells located at the edge (orange) and cells located at the centre (purple) at 48 hours of LIF withdrawal, with the protein levels Phos-Pol II being more homogeneously distributed than p53 and BRACHYURY.

Furthermore, we have included a UMAP analysis of individual cells during 0-48 hours of LIF withdrawal. This analysis shows that between 0 and 12 hours of LIF withdrawal there are the largest changes in gene expression levels, with 24 and 48 hours of LIF withdrawal being the most similar. See Figure 5g, pasted here for convenience:

We have updated the main text to read (page 12, line 357-361):

“When we project the data on a 2D UMAP, we observe the most prominent separation of cellular expression profiles occurring between 0-12 hours post LIF withdrawal (Fig. 5g). Furthermore, while there is some separation in cells located at the edge and centre, this separation is subtle, and predominantly visible at 48 hours post LIF withdrawal (Fig. 5g).”

Reviewer 4, comment 3:

All over the manuscript, the number of experiments or colonies used in each figure is not always clear. For instance in fig 4abc. Is data presented for one colony only? This might be OK for panels a and b but for panel c it might be stronger to show localization data averaged over many colonies and full biological replicates. Same for fig5cd; is this data from 1 colony or an average of many colonies? The number of samples should be stated in figure legends.

Response

We have now adjusted Figure 4c to show the data across 3 biological replicates. Furthermore, we have replaced the data in Figure 5d-f as well as Supplementary Fig. 26 (previously Figure 5f) to include biological replicates as well. Unfortunately, it would be too time-consuming to perform the sequential imaging on multiple micropatterns per biological replicate. The imaging of one micropattern already takes 8 hours over 5 rehybridization rounds and generates around 6000 images. Therefore, we prioritised multiple biological replicates over multiple micropatterns per replicate. We have now added the number of replicates, micropatterns per replicate, cells per

micropattern and positions from each micropattern (if appropriate) to each figure legend (i.e., Fig 3b-d, g; Fig 4c-g; Fig 5d-g; Fig 6b-c, f, i-j; Fig 7e-h).

Reviewer 4 comments related to Reviewer 3's comments

3a) OK it's good to analyse 4 biological replicates but it would be better to be a bit more precise. What is meant by biological replicates? How many colonies were considered by replicates. This info should be in the legend of the figure.

Response:

We thank the reviewer for pointing this out and have now explicitly labelled how many replicates, how many micropatterns per replicate, how many cells per micropattern and how many positions from the micropatterns (if appropriate) were analysed in the associated figure legend (i.e., Fig 3b-d, g; Fig 4c-g; Fig 5d-g; Fig 6b-c, f, i-j; Fig 7e-h).

3b) The statement “lineage-associated proteins Brachyury and GATA6 are mostly enriched in cells at the edge of the micropattern (Fig. 4e and Supplementary Fig. 25a), as are the pluripotency proteins Nanog and Sox2” is puzzling to me. Cells are both pluripotent and differentiated at the edge of the colony? Is this over expression in the same cells? What are those cells then? The description of marker co-expression should be done at the single cell level (see my comment in intro)

Response:

We thank the reviewer for raising this point. When looking specifically at the single cell levels of BRACHYURY and GATA6 compared to NANOG and SOX2 in cells at the edge of the micropattern, we do not see a positive correlation between the protein levels of individual cells (see below). Notably, we do not believe that these cells are differentiated into any specific germ layers, because 48 hours of LIF withdrawal is not enough to see the emergence of those differentiated cells. Therefore, it is not surprising that while many cells are not expressing any NANOG or SOX2, others are still highly heterogeneous in their expression levels. Yet it is not necessarily the same cells expressing high levels of pluripotency as well as lineage-associated proteins. We have now included this analysis in Supplementary Figure 25e, pasted here for convenience:

We have added a sentence in the main text reflecting this new analysis (see page9, line235-236):

“Yet, *BRACHYURY* and *GATA6* are not necessarily highly expressed in the same cells as *Nanog* and *SOX2* (Supplementary Fig. 25e).”

9a) OK but proper negative controls could have been added. They are not always possible but for some markers they can be easy. For instance for phospho-SMAD2 using the SB43 inhibitor as a negative control and activin stimulation as a positive one give a lower and upper bound of the signal.

Response:

We thank the reviewer for the suggestion and have now quantified the lower and upper bound of Phos-SMAD1 and Phos-SMAD2 signal. Specifically, as a negative control, we tested Phos-SMAD proteins in two different cell lines where they have previously been reported to be lowly expressed (i.e., Phos-SMAD is low expressed in MCF7 and Phos-SMAD1 is low expressed in NIH3T3)^{1, 2}. Conversely, in both cell lines total-SMAD and SMAD1 are highly expressed. We have pasted the literature evidence below:

As the western blots above show, the level of Phos-SMAD is undetectable in MCF7 cells (left) and Phos-SMAD1 is very low in the NIH3T3 cells (right). We have marked in red the specific bands that show this.

We had already performed detection of both Phos-SMAD1 and Phos-SMAD2 in these two cell-types (i, bottom and middle) using ARTseq-FISH and compared it to the signal observed in mESCs (i, top). We have now included additional quantification of the signal (j), which demonstrates that in NIH3T3 cells—where we expect no expression of Phos-SMAD1²—we indeed observe Phos-SMAD1 levels comparable to the ARTseq-FISH negative control (see NIH3T3 Phos-SMAD1 compared to Neg ctrl_AT). Therefore, ARTseq-FISH appears capable of detecting a true lack of Phos-SMAD1 protein.

Similarly, for Phos-SMAD2 detection in MCF7 and NIH3T3 cells are also expected to be very low^{2, 3}. ARTseq-FISH analysis shows that in these two cell lines, the detection of Phos-SMAD2 is only slightly higher than the ARTseq-FISH negative control (see MCF7 and NIH3T3 Phos-SMAD2 compared to Neg ctrl_Tm). Therefore, ARTseq-FISH also appears to accurately detect a true lack of Phos-SMAD2.

Lastly, from the western blot results above^{2, 3} total SMAD protein and SMAD1 are expected to be abundantly expressed in MCF and NIH3T3 respectively, which is also reflected in the ARTseq-FISH analysis. In fact, the detection of SMAD1 is almost 3-fold higher in NIH3T3 cells compared to mESCs, indicating that the upper limit of detection of SMAD1 is >3-fold higher than what we detect in mESCs.

This new analysis is now included in Supplementary Fig. 3: i-j, pasted here for convenience:

Low expressed Phos-SMAD in MCF7 and NIH3T3 compared to mESCs. i, (Left) Example images of Phos-SMAD1, Phos-SMAD2 and SMAD1 proteins detected by ARTseq-FISH in MCF7 and NIH3T3 cells. Images were set as the same max/min grayvalue. Scale bar, 20 μ m. (Right) Western blot detection of p-Smad, PSAMD1/5 and SMAD proteins in MCF7 and NIH3T3 cells respectively from previous studies^{2, 3}. **j,** Quantification of spots per cell of Phos-SMAD1, SMAD1, and Phos-SMAD2 in mESC, MCF7 and NIH3T3 cell lines. Neg ctrl_AT is negative control in the green channel with ATTO488 fluorophore. Neg ctrl_Tm is negative control in the red channel with TAMRA fluorophore. Line in the boxenplot denoting the median.

9b) Isn't it puzzling that IF and ArtseqFISH data don't agree in fig 5f? significance of the differences at 0 and 48hr should be assessed. I don't agree that the difference is subtle for artseqfish at 0 vs 48h while it doesn't seem significant in IF

Response:

We do believe that the IF and the ARTseq-FISH data agree. Previously we had plotted the error bars as 95% confidence intervals of >100 cells for each position analysed by IF. Therefore, the error bars were perhaps not the best representation of the distribution of the single-cell data, for which we apologize. Instead, we now plotted the data as boxenplots, which better represents the range of expression levels observed at a single cell level with both IF and ARTseq-FISH. While the difference between 0 and 48 hours is not significant for both the IF and the ARTseq-FISH data, similar trends are observable. Notably, in response to Reviewer 4, comment 3, the ARTseq-FISH data now includes single-cell data from 3 biological replicates. Due to space constraints caused by the addition of new Figure 5f (see Reviewer 4, comment 2), we have now moved this figure to Supplementary Fig. 26, pasted here for convenience.

OCT4 protein detected by IF and ARTseq-FISH. Boxenplot shows the normalised intensity of OCT4 detected by immunofluorescence at the edge (top left) and centre (bottom left) of the micropattern at 0 hr and 48 hr post LIF withdrawal. Boxenplot shows the normalised expression of OCT4 protein detected by ARTseq-FISH at the edge (top right) and centre (bottom right) of the micropattern at 0 hr and 48 hr post LIF withdrawal. Line in the boxenplot denotes the median.

10) I agree with this concern and I think the solution chosen is not fully satisfying given the density of the tissue. A segmentation based on plasma membrane would be better in that case but I concede that this is more difficult to achieve. Maybe keeping only the signal overlapping with nuclei would minimize the possibility of assignment mistake?

Response:

We would like to point out that other comparable techniques use similar segmentation methods⁴⁻⁹. Yet, we have now performed an additional analysis where we analysed only signal overlapping with the nucleus, using a low threshold for DAPI detection. This causes the nuclear mask to be larger than the true nucleus. We compared the spot count per cell of all targets when using our original segmentation method to only the nuclear mask. Since mESCs have a very large nucleus, we found that there is a very strong correlation indicating that there is not much error introduced by our segmentation method. This analysis is now included in SN2 Fig. 23, pasted here for convenience:

SN2 Fig. 23. Correlation calculated between the average spots per cell in the nucleus and the whole cell. Data includes 3 positions taken from 3 separate images.

REFERENCES

1. de Silva, H. C.; Firth, S. M.; Twigg, S. M.; Baxter, R. C., Interaction Between IGF Binding Protein-3 and TGF β in the Regulation of Adipocyte Differentiation. *Endocrinology* **2012**, *153* (10), 4799-4807.
2. Miller, D. S. J.; Schmierer, B.; Hill, C. S., TGF- β family ligands exhibit distinct signalling dynamics that are driven by receptor localisation. *J Cell Sci* **2019**, *132* (14).
3. Li, L.; Qi, L.; Liang, Z.; Song, W.; Liu, Y.; Wang, Y.; Sun, B.; Zhang, B.; Cao, W., Transforming growth factor- β 1 induces EMT by the transactivation of epidermal growth factor signaling through HA/CD44 in lung and breast cancer cells. *Int J Mol Med* **2015**, *36* (1), 113-22.
4. Chen, K. H.; Boettiger, A. N.; Moffitt, J. R.; Wang, S.; Zhuang, X., Spatially resolved, highly multiplexed RNA profiling in single cells. *Science* **2015**, *348* (6233), aaa6090.
5. Sountoulidis, A.; Lontos, A.; Nguyen, H. P.; Firsova, A. B.; Fysikopoulos, A.; Qian, X.; Seeger, W.; Sundström, E.; Nilsson, M.; Samakovlis, C., SCRINSHOT enables spatial mapping of cell states in tissue sections with single-cell resolution. *PLOS Biology* **2020**, *18* (11), e3000675.
6. Saka, S. K.; Wang, Y.; Kishi, J. Y.; Zhu, A.; Zeng, Y.; Xie, W.; Kirli, K.; Yapp, C.; Cicconet, M.; Beliveau, B. J.; Lapan, S. W.; Yin, S.; Lin, M.; Boyden, E. S.; Kaeser, P. S.; Pihan, G.; Church, G. M.; Yin, P., Immuno-SABER enables highly multiplexed and amplified protein imaging in tissues. *Nat Biotechnol* **2019**, *37* (9), 1080-1090.
7. Gyllborg, D.; Langseth, C. M.; Qian, X.; Choi, E.; Salas, S. M.; Hilscher, M. M.; Lein, E. S.; Nilsson, M., Hybridization-based in situ sequencing (HybISS) for spatially resolved transcriptomics in human and mouse brain tissue. *Nucleic Acids Research* **2020**, *48* (19), e112-e112.
8. Lee, H.; Marco Salas, S.; Gyllborg, D.; Nilsson, M., Direct RNA targeted in situ sequencing for transcriptomic profiling in tissue. *Scientific Reports* **2022**, *12* (1), 7976.
9. Goh, J. J. L.; Chou, N.; Seow, W. Y.; Ha, N.; Cheng, C. P. P.; Chang, Y.-C.; Zhao, Z. W.; Chen, K. H., Highly specific multiplexed RNA imaging in tissues with split-FISH. *Nature Methods* **2020**, *17* (7), 689-693.

Reviewer #4 (Remarks to the Author):

Hu et al. in this revised version of their manuscript have provided clarification that addressed the points previously raised in a satisfactory manner. The article now clearly shows the potential of the ARTseq methods to analyze how gene expression varies with cell position in a colony. In particular, I agree that removing attempts at description of cell identity and only describing the expression of particular proteins or mRNAs addressed the confusion about biology of the system.

I remained convinced that ARTseq is a great technology to assess cell identity and it is a bit underwhelming that this potential is not fully illustrated here as the study is focused at a time point (48h after LIF removal) when cell identities are not really marked. Analyzing later time points when germ layers are clearly defined would make an interesting follow-up study!

Minor points:

in several figure legends it is mentioned "UMAP analysis of individual cells" it would be useful to be more explicit about what is used in the UMAP, e.g. : UMAP analysis of expression levels of xx markers of individual cells...

Terminology: I wouldn't use "expression" to describe pSMAD1 levels. As it represents the level of modification (phosphorylation) of SMAD1. SMAD1 could be expressed but not phosphorylated.

Reviewer #4 (Remarks to the Author):

Hu et al. in this revised version of their manuscript have provided clarification that addressed the points previously raised in a satisfactory manner. The article now clearly shows the potential of the ARTseq methods to analyze how gene expression varies with cell position in a colony. In particular, I agree that removing attempts at description of cell identity and only describing the expression of particular proteins or mRNAs addressed the confusion about biology of the system.

I remained convinced that ARTseq is a great technology to assess cell identity and it is a bit underwhelming that this potential is not fully illustrated here as the study is focused at a time point (48h after LIF removal) when cell identities are not really marked. Analyzing later time points when germ layers are clearly defined would make an interesting follow-up study!

Minor points:

in several figure legends it is mentioned “UMAP analysis of individual cells” it would be useful to be more explicit about what is used in the UMAP, e.g. : UMAP analysis of expression levels of xx markers of individual cells...

Response:

We have now changed this to read:

“UMAP analysis of expression levels of 57 markers of individual cells ..” (Figure 4e)

“UMAP clustering of expression levels of 57 markers of the individual cells ..” (Figure 5gi)

“UMAP clustering of expression levels of 57 markers of single cells ..” (Figure 5gii)

The lowest abundant markers (i.e. those that were rarely detected) were left out of this analysis. The data is provided in Source Data.xlsx

Terminology: I wouldn't use “expression” to describe pSMAD1 levels. As it represents the level of modification (phosphorylation) of SMAD1. SMAD1 could be expressed but not phosphorylated.

Response:

We have now changed the term “expression” to “levels”, “abundance”, or “number of spots” when referring to phosphorylated proteins in the main text.